# A hollow TFG condensate spatially compartmentalizes the early secretory pathway

Savannah M. Bogus [1,3], William R. Wegeng [1,3], Miguel Ruiz[1], Sindy R. Chavez[1], Samantha N. Cheung [1], Khalid S. M. Noori[1], Ingrid R. Niesman [2] & Andreas M. Ernst [1] ✉

In the early secretory pathway, endoplasmic reticulum (ER) and Golgi membranes form a nearly spherical interface. In this ribosome-excluding zone, bidirectional transport of cargo coincides with a spatial segregation of anterograde and retrograde carriers by an unknown mechanism. We show that at physiological conditions, the Trk-fused gene (TFG) self-organizes to form a hollow, anisotropic condensate that matches the dimensions of the ER–Golgi interface and is dynamically regulated across the cell cycle. Regularly spaced hydrophobic residues in TFG control the condensation mechanism and result in a porous condensate surface. We find that TFG condensates act as a molecular sieve capable of allowing access of anterograde coats (COPII) to the condensate interior while restricting retrograde coats (COPI). We propose that a hollow TFG condensate structures the ER–Golgi interface to create a diffusion-limited space for anterograde transport. We further propose that TFG condensates optimize membrane flux by insulating secretory carriers in their lumen from retrograde carriers outside TFG cages.

In the early secretory pathway, the interface formed between the endoplasmic reticulum (ER) and the Golgi stack collaborates in the secretion of biosynthetic cargo[1]. Specialized domains of the ER, termed ER exit sites (ERES), form metastable contacts with the *cis* face of the Golgi apparatus. Morphologically, the ER–Golgi interface is defined by concave membranes at ERES and the *cis*-Golgi, resulting in a nearly spherical, ribosome-excluding zone < 500 nm in diameter[2,3]. The ER–Golgi interface functions as the primary 'valve' of the secretory pathway: in a typical human cell, ~30% of the proteome, as well as membrane lipids are routed through this interface, corresponding to ~50% of the ER volume emptying into this junction every 40 min, while simultaneously retrieving ~90% of the membrane and recycling it back to the ER[1,4]. The exchange of material at the interface is accomplished by bidirectional transport via COPII and COPI complexes, which deform the membrane to produce vesicular and tubular carriers. COPII carriers originate in the ER and are directed towards the Golgi

(anterograde), while COPI carriers operate in the opposite (retrograde) direction[5]. The small, submicron dimension of the interface thus poses the problem of clashes between carriers headed in the opposite direction, which would result in content mixing and eventually loss of compartmental identity. Several decades ago, immuno-EM approaches revealed a strict spatial separation of COPI and COPII machineries, with COPII carriers concentrated in the center of the interface while COPI coats were segregated towards the periphery[6–9]. How this spatial separation is achieved remains unexplained.

Recently, evidence has emerged suggesting condensation of the ER–Golgi interface proteins TANGO1 at ERES, Sec16 and Trk-fused gene (TFG) within the interface, and GM130 at the *cis*-Golgi[10–18], suggesting that the early secretory pathway is structured by self-organizing protein collectives. Among these proteins, Sec16 and TFG have been proposed as candidates capable of structuring the space between ERES and the *cis*-Golgi. Sec16, a peripheral membrane protein, was

[1]Department of Cell and Developmental Biology, School of Biological Sciences, University of California San Diego, La Jolla, CA, USA. [2]Department of Biology, San Diego State University, San Diego, CA, USA. [3]These authors contributed equally: Savannah M. Bogus, William R. Wegeng. ✉e-mail: aernst@ucsd.edu

found to co-condense with TANGO1[18], which is a transmembrane component of ERES[12]. Importantly, the heterogeneous Sec16 condensates were found to impact cargo flux through the early secretory pathway. Silencing of TFG in *C. elegans* strongly impacted ER–Golgi interface morphology and resulted in both diminished Golgi membranes and mislocalized COPII carriers[10], which is paralleled by results obtained from HeLa cells that support a role for TFG in organizing ERES and anterograde carriers into larger, coherent structures[19]. Similar to Sec16, TFG was shown to critically impact the amount of cargo exported from the ER[10], but not its export kinetics, while secretion of bulky cargo (i.e., collagen) was significantly impaired[19]. When recombinant TFG purified from bacteria was subjected to high concentrations of potassium acetate (KOAc), it precipitated out of solution to form amorphous, reversible aggregates[11]. These data prompted the speculation that TFG might form a biomolecular condensate capable of organizing the ER–Golgi interface as 'molecular glue', with additional functions in uncoating of anterograde carriers[13,14]. How a putative TFG condensate could contribute to the spatial segregation of anterograde and retrograde carriers at the interface remained elusive.

## Results and discussion

### Recombinant TFG assembles into anisotropic, hollow condensates in vitro

We set out to test whether a protein condensate at the ER–Golgi interface simultaneously defined its structure and controlled the organization of bidirectional traffic. If the intrinsically disordered proteins Sec16 and TFG were responsible for structuring the spherical ER–Golgi interface, they should localize between ERES and *cis*-Golgi membranes. To localize TFG and Sec16 at individual ERES/Golgi pairs, we employed the microtubule-destabilizing agent nocodazole to 'unlink' the convoluted Golgi 'ribbon'[20]. Next, we co-transfected cells with mGFP-Sec16L and FLAG-TFG-SNAP, and immunolabeled *cis*-Golgi (GM130) and ERES (TANGO1) markers. Localization analysis indicated that TFG was present precisely between ERES and the *cis*-Golgi, whereas Sec16 appeared restricted to the ERES membranes (Fig. 1a and Supplementary Fig. 1a, b).

Due to TFG populating the interface, we next set out to test if recombinant TFG was capable of forming condensates in vitro. For this purpose, we generated pure (>95%) recombinant TFG from human suspension cells (Fig. 1c and Supplementary Fig. 2a). When

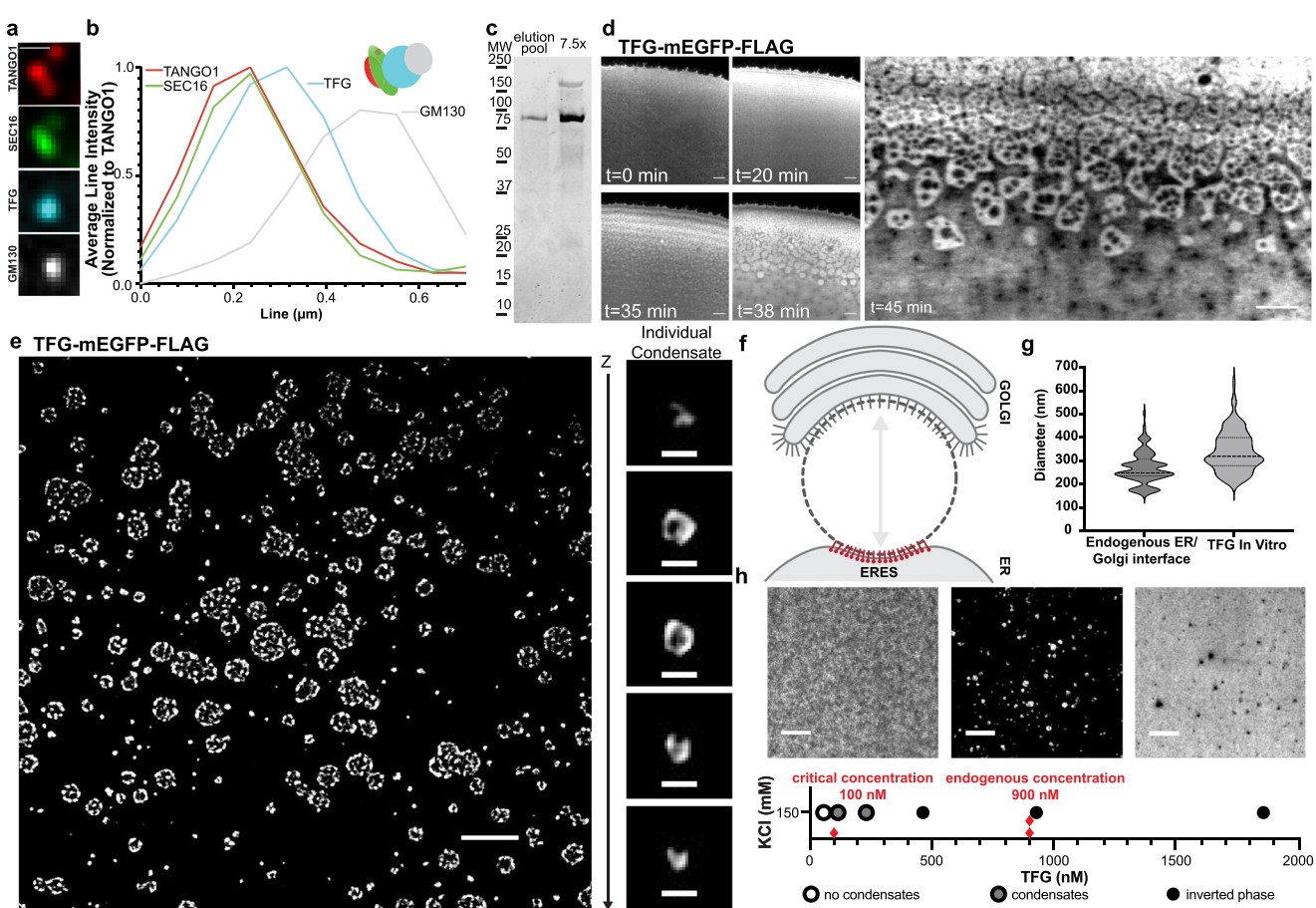

**Fig. 1 | Recombinant TFG assembles into anisotropic, hollow condensates in vitro. a** Representative 4-color micrograph of an individual ERES/*cis*-Golgi unit. HeLa cells were transfected with mGFP-Sec16L (green), FLAG-TFG-SNAP (cyan) (labeled with SNAP-Cell 647-SiR), and immunostained for endogenous GM130 (gray) and TANGO1 (red). Scale bar 1 μm. **b** Fluorescence intensity line scan of 4-color ERES/*cis*-Golgi unit (averaged and normalized to TANGO1, n = 4). Schematic of localizations detected in the representative unit (**a**) is given. **c** Representative purity of peak TFG elutions from concentration/salt exchange (Coomassie staining). **d** Confocal time series of increasing concentration of TFG-mEGFP-FLAG at the edge of an evaporating 10 μL sessile droplet, 150 mM KCl. Scale bar 5 μm. Starting concentration -100 nM. **e** TFG-mEGFP-FLAG at 430 nM spontaneously assembles into 'sponge-like', anisotropic condensates (HEPES/KOH pH 7.3; 150 mM KCl; 20% (v/v) PEG 8 kDa; confocal microscopy). Scale bar 5 μm. Individual TFG condensate, Z-step 125 nm. Scale bar 500 nm. **f** Schematic depicting the ER–Golgi interface (targets for immunolabeling at ERES: red; *cis*-Golgi: grey. **g** ERES to *cis*-Golgi distance measured between immunolabeled GM130 and TANGO1, mean = 269, n = 120, vs diameter of individual hollow TFG condensates in vitro, mean = 341, n = 175. **h** Determination of the saturation concentration required for the formation of hollow TFG condensates (in the absence of crowding agents). Representative micrographs depicting the distribution of TFG below and above the saturation concentration of 100 nM. Higher concentrations of TFG yield inverted phases (right micrograph). Dilute phase (left) shown at <5 nM, dense phase (middle) shown at 200 nM, inverted phase (right) shown at 6.4 μM. Red diamonds indicate saturation concentration and average cellular concentration of TFG. Scale bars 5 μm. Source data are provided as a Source Data file.

characterizing recombinant TFG in supersaturated solutions[15,21], we concentrated the protein by sessile droplet evaporation and observed phase-separation into condensates with an anisotropic structure, forming ~500 nm voids within the dense phase that paralleled the structure of condensates formed upon overexpression in cells (Fig. 1d). The anisotropic distribution of protein with these condensates developed from an initially isotropic distribution of TFG (Fig. 1d). These data support the sufficiency of TFG to form anisotropic condensates independently of other components.

We next tested whether formation of TFG condensates is promoted or inhibited by a crowding agent, PEG, which is frequently employed in vitro to mimic macromolecular crowding experienced by proteins in the cytosol[22]. When recombinant TFG was exposed to PEG (8 kDa, 20% (v/v)), we observed rapid induction of TFG condensates with a broad size distribution (Fig. 1e). TFG formed both micron-sized, sponge-like condensates, as well as condensates < 500 nm that appeared spherical, closely matching the dimensions of the ER–Golgi interface obtained from immunolabeling ERES and cis-Golgi markers (endogenous ER–Golgi interface: 270 nm ± 65 (Fig. 1f); TFG condensates in vitro: 341 nm ± 87; Fig. 1g). The submicron condensates exhibited a void in their center, resembling the shape of a hollow sphere (Fig. 1e). Notably, the formation of anisotropic, hollow individual condensates did not result from the presence of nucleic acids in the condensate core (Supplementary Fig. 2c). These data suggested that the sponge-like appearance of larger condensates resulted from coalescence of individual < 500 nm hollow condensates. Thus, TFG self-organizes to form submicron-sized, hollow spherical condensates that could explain the spherical void at the ER–Golgi interface. The micron-sized 'sponge-like' condensates may also represent physiologically relevant structures, as ERES are clustered under the Golgi ribbon[2], a process that would be facilitated by the coalescence of individual submicron condensates. We next set out to test whether the submicron condensates would form under physiological conditions and concentrations (Fig. 1h). We concentrated recombinant TFG of defined starting concentrations stepwise to determine the saturation concentration required for condensate formation. Condensation was observed already at concentrations of ~100 nM, which is below the saturation concentration typically reported for intrinsically disordered proteins to undergo liquid-liquid phase separation (LLPS; 10–100 μM[23–26]). Condensation at such a low saturation concentration is consistent with specific interactions among TFG molecules. When approaching concentrations of ~1 μM, we observed the formation of inverted phases with submicron aqueous inclusions. As the average cellular concentration of TFG is ~900 nM[27], these observations support the ability of TFG to form condensates under physiological conditions.

To confirm the physiological relevance of TFG as a cytosolic extension of ERES and its role in controlling flux of cargo from the ER, we employed the retention using selective hooks (RUSH[28]) system with a KDEL-Str hook and the model cargo SBP-EGFP-ECadherin to compare the rate of cargo export in wild-type cells and cells in which expression of TFG was silenced. The absence of TFG significantly diminished the rate of cargo clearance from the ER and resulted in compromised cell growth (Supplementary Fig. 3d, b). This paralleled prior data obtained for the temperature-sensitive model cargo ts045-VSV-G, collagen, and TFG hepatocyte knockout mice[10,19,29].

### Endogenous tagging and overexpression confirm assembly of TFG into anisotropic condensates

Next, we set out to test whether the hollow TFG condensates observed in vitro could be detected in cells at physiological concentrations. Employing CRISPR/Cas9 in HeLa, we endogenously tagged TFG at its C-terminus with mClover-FLAG, and validated homozygous clones by sequencing and Western blot (Supplementary Fig. 5a). When performing live-cell microscopy of TFG::mClover-FLAG, we observed discrete foci reminiscent of condensates throughout the cell, as well as a diffuse

distribution in the cytosol supportive of the presence of a two-phase coexistence (i.e., dilute and dense TFG phases) (Fig. 2a and Supplementary Fig. 5b). In the dense TFG foci, the distribution appeared to be anisotropic, paralleling the results obtained in vitro (Fig. 1e). Furthermore, the average diameter of TFG condensates in vivo closely mirrored the dimensions observed in vitro (Fig. 2b; TFG condensates in vitro: 341 nm ± 87; TFG condensates in vivo: 363 nm ± 78), strongly supporting that TFG can self-organize into hollow condensates that match the dimensions of the ER–Golgi interface in live cells and at physiological concentrations and conditions. When photobleaching entire endogenous TFG condensates in vivo, we observed a low extent of fluorescence recovery (FRAP; Fig. 2c). Analysis of the mobility of TFG in endogenous condensates (partial FRAP) could not be robustly performed due to their submicron dimension and dynamics.

To further characterize the dynamics of TFG condensates, we overexpressed TFG in HeLa cells. TFG initially exhibited a dispersed localization throughout the cytosol, and suddenly began to condense into dynamic sub-micron foci (Supplementary Fig. 4a). These foci increased in size over the scale of minutes and fused to form structures several microns large. Condensation of TFG upon overexpression did not depend on the mEGFP tag, since it was observed with diverse genetically encoded fluorescent (SNAP, mCherry) and epitope tags (Supplementary Fig. 4b). Furthermore, TFG condensates efficiently recruited mGFP-Sec16L to their surface (Supplementary Fig. 4c), paralleling reports of a specific interaction between Sec16 and TFG[10]. To gain further insight into the distribution of TFG within condensates, we employed both stimulated emission depletion (STED) and 3D-structured illumination (3D-SIM) super-resolution microscopy (Fig. 2d). TFG-mEGFP-FLAG condensates neither exhibited smooth surfaces nor a high degree of sphericity (Fig. 2d and Supplementary Fig. 6b), but rather appeared sponge-like, paralleling our results obtained from supersaturated TFG and upon crowding in vitro. Interestingly, the voids within the dense condensate phase exhibited diameters that resemble the diameter of the ER–Golgi interface, mirroring the in vitro observation that sponge-like condensates result from coalescence of individual hollow TFG condensates (Fig. 1e). FRAP experiments supported a low mobility of TFG within condensates, as well as a low extent of exchange of TFG between the dense condensate phase and cytosol (partial FRAP: $n = 33$; full FRAP: $n = 30$; Fig. 2e, f) indicating viscoelastic material properties, with approximately 70% of TFG molecules' diffusion being significantly confined within the condensate dense phase. However, micron-sized TFG condensates exhibited 'capillary bridges' during fusion and relaxed back into overall spherical geometries after fusion, which together indicate the presence of surface tension and the formation of flexible TFG collectives (Fig. 2g, h). When performing live-cell confocal imaging of large TFG condensates, we observed diffraction-limited heterogeneities supportive of an anisotropic distribution of TFG within the condensate dense phase (Supplementary Fig. 6b). Our data thus far highlight TFG's propensity to self-organize into ER–Golgi interface-sized, anisotropic (hollow) condensates. However, the low mobility of TFG within the dense phase raised the question of how these structures could be regulated and disassembled.

### TFG condensation is regulated by phosphorylation

Multiple proteins that populate the early secretory pathway are regulated by phosphorylation[18,30,31], leading to the dispersal of ERES and Golgi membranes upon mitotic entry, and followed by a rapid reassembly upon mitotic exit[32]. TFG contains several putative serine/threonine, as well as tyrosine phosphorylation sites that are dispersed across its entire length (Fig. 3c), suggesting that condensates could be regulated by phosphorylation as well. We thus set out to test whether TFG would disassemble during mitosis in vivo and in vitro. Capitalizing on the endogenously tagged TFG::mClover-FLAG HeLa cell line, we identified mitotic cells via Hoechst staining (Fig. 3a). Strikingly, TFG condensates could not be detected in mitotic cells, instead exhibiting a

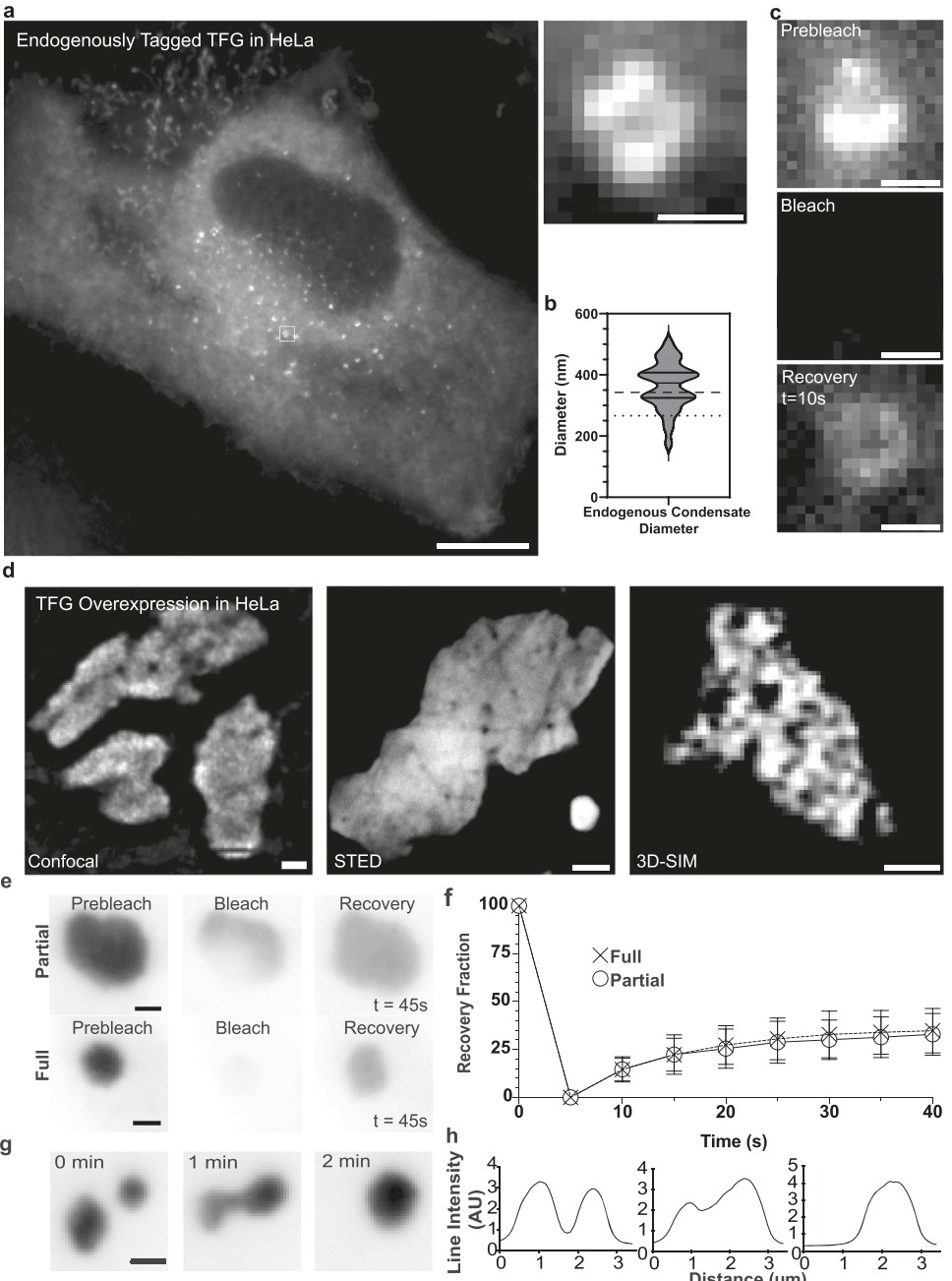

**Fig. 2 | Endogenous tagging and overexpression confirm assembly of TFG into anisotropic condensates that disassemble during mitosis. a** Representative widefield micrograph of HeLa cells with homozygous endogenously tagged TFG::mClover-FLAG. Scale bar 10 μm. Magnified micrograph (dashed box in cell overview): individual TFG condensate. Scale bar 500 nm. **b** Quantification of the diameter of endogenous TFG condensates = 30. The dashed line represents the average of in vitro condensate diameter (341 nm ± 87) and the dotted line represents the average of endogenous ER/Golgi interface diameter (269 nm), for reference. **c** Qualitative fluorescence recovery after photobleaching (FRAP) of a TFG condensate. Scale bar 500 nm. **d** Left: live-cell confocal microscopy of large individual condensates within HeLa cells transfected with TFG-mEGFP-FLAG. Scale bar

5 μm. Middle: STED micrograph of large individual condensates within HeLa cells transfected with TFG-mEGFP-FLAG. Scale bar 2 μm. Right: 3D-structured illumination microscopy (3D-SIM) of large individual condensates within HeLa cells transfected with TFG-tGFP. Scale bar 500 nm. **e** Representative micrographs of partial and full FRAP experiments on overexpressed TFG condensates in HeLa cells. **f** Quantification of average condensate recovery rates for partial (*n* = 33) and full (*n* = 30) FRAP; normalized to pre-bleach (=100) and post-bleach intensity (=0). Error bars represent standard deviation. **g** Representative live-cell imaging of TFG condensate fusion dynamics. **h** Line scan of condensate depicted in (**g**). Source data are provided as a Source Data file.

dispersed localization within the cytosol. We next set out to validate this result in vitro by synchronizing Expi293F cells prior to transfection and overexpression of TFG. Mirroring the results obtained from endogenously tagged TFG, recombinant TFG purified from mitotic cells failed to form condensates, while protein purified from interphase cells efficiently assembled into hollow condensates (Fig. 3b).

These data support that TFG is dynamically regulated across the cell cycle along with other structural proteins in the early secretory pathway. To test whether this effect could be attributed to phosphorylation, as has been shown in the ERES protein TANGO1[31], we treated recombinant TFG purified from mitotic cells with lambda protein phosphatase (Fig. 3b and Supplementary Fig. 7a). Upon phosphatase

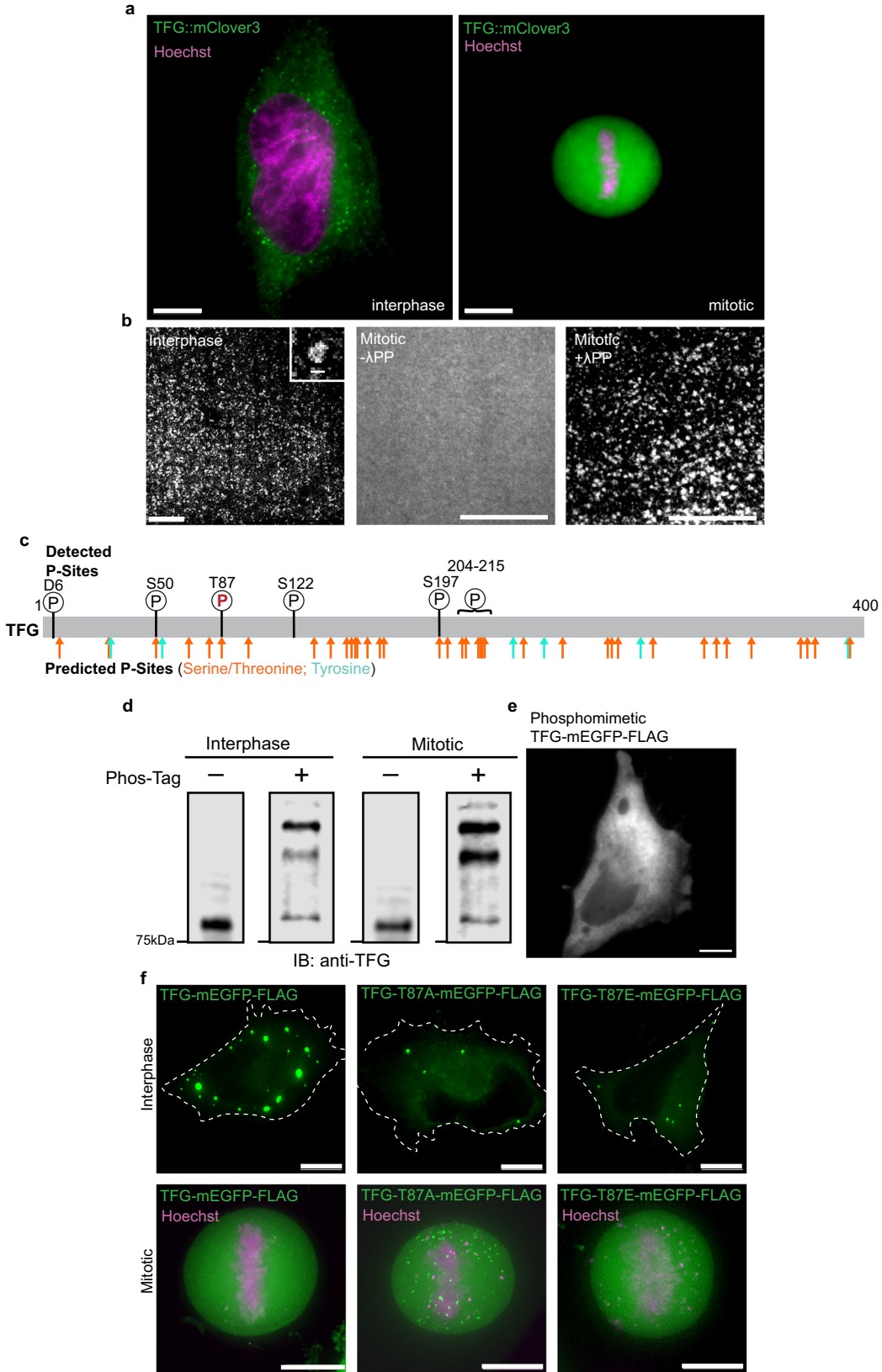

treatment, the ability to undergo phase separation was restored, and TFG (162–240) assembled readily into anisotropic condensates. These data suggest that phosphorylation sites across the protein lead to inter- and intra-molecular repulsion and thus inhibition of condensate formation during mitosis, with dephosphorylation allowing rapid reassembly of TFG into hollow spheres upon mitotic exit.

To confirm whether repulsion at phosphorylation sites would be sufficient to inhibit TFG condensation, we overexpressed a phospho-mimetic TFG-mEGFP-FLAG construct in which all predicted phosphorylation sites were mutated to a negatively charged residue (i.e., serine mutated to aspartic acid and threonine mutated to glutamic acid) (Fig. 3e and Supplementary Fig. 7b). Phosphomimetic TFG failed

**Fig. 3 | TFG is regulated by phosphorylation across the cell cycle.**
**a** Representative micrograph of interphase (left) and mitotic (right) HeLa cells with endogenously tagged TFG. Cells were stained with Hoechst (magenta). Scale bar 10 μm. **b** Micrographs of recombinant TFG-162-240-mEGFP-FLAG purified from synchronized interphase or mitotic Expi293F cells. Mitosis-derived protein was treated with lambda protein phosphatase (λPP) where indicated. Scale bar 10 μm; inset 500 nm **c** Schematic depicting predicted phosphorylation sites within TFG (bottom): serine/threonine, orange; tyrosine, cyan, and depicting confidently identified phosphorylation sites found via mass spectrometry in TFG (top) with mitotic phosphorylation site depicted with red/bolded 'P'. Putative phosphorylation sites were predicted by NetPhos-3.0. **d** Representative anti-TFG Western blot of PhosTag gel depicting purified recombinant TFG-mEGFP-FLAG derived from synchronized cells in interphase (I) or mitosis (M) as indicated. Arrows indicate species

that are less abundant (lowest arrow) or more abundant (upper arrows). Blank lane between each condition was cropped. **e** Representative micrograph of phospho-mimetic TFG-mEGFP-FLAG 24 h post-transfection in HeLa cells. Widefield microscopy, Z-slice, scale bar 10 μm. **f** Representative widefield deconvolution micrographs of interphase (upper) and mitotic (lower) HeLa cells overexpressing TFG-mEGFP-FLAG (left), TFG-T87A-mEGFP-FLAG (phosphor-dead) (middle), or TFG-T87E-mEGFP-FLAG (phosphomimetic) (right). Cells were stained with Hoechst (magenta). Scale bar 10 μm. 90% of mitotic cells expressing wild-type TFG ($n = 24$) or TFG-T87E ($n = 10$) were able to fully dissolve foci, in contrast to 65% in T87A-expressing cells ($n = 26$). At least three independent experiments were performed to obtain the replicates for each condition. Source data are provided as a Source Data file.

to form condensates when overexpressed in HeLa cells, again indicating that repulsion via negative charges would be sufficient to inhibit TFG condensation. Next, we were interested in testing whether changes in phosphorylation occur throughout the cell cycle. We employed recombinant TFG-mEGFP-FLAG obtained from synchronized cells that were either in interphase or mitotic, and analyzed them in a Phos-tag gel followed by Western blotting for TFG (Fig. 3d). Several bands are detected for both samples that are absent in control gels, indicating that TFG is phosphorylated across the cell cycle. However, in the interphase-derived sample, we found a high fraction of molecular phospho-TFG species that migrated faster, indicating *less* phosphorylation, whereas mitotic TFG exhibited a high fraction of molecular phospho-species that ran slower, indicating *more* phosphorylation. Based on these data, we conclude that TFG is a phosphoprotein across the cell cycle, but becomes hyperphosphorylated during mitosis. In order to elucidate this further, we performed mass spectrometry on recombinant TFG purified from synchronized cells. We found four high-confidence phosphorylation sites that were present in samples derived from both interphase and mitosis, indicating that TFG is a phosphoprotein throughout the cell cycle: S50, S122, S197, and interestingly D6, which is a non-canonical phosphorylation site (Fig. 3c). Importantly, we also identified T87 as a mitosis-specific phosphosite which was not detected in interphase samples although corresponding peptides that were unmodified were detected. Notably, phosphorylation was consistently detected between aa 204–215, however, limitations in mass spectrometry prevent accurate localization or quantification within this region, as it contains a cluster of serine and threonine residues that lack internal arginine or lysine residues to enable cleavage by trypsin. It is furthermore likely, supported by the shifts observed in Phos-tag gels, that the detected high-confidence sites are an underestimation of all existing phosphosites in TFG during interphase and mitosis. Furthermore, the serine-rich region might be important for cell cycle-mediated regulation of TFG as it is also present in TFG-162–240-mEGFP-FLAG, for which mitotic protein had rapidly assembled into condensates upon dephosphorylation (Fig. 3b). Concordantly, when overexpressed in HeLa cells, wildtype TFG and phosphomimetic TFG-T87E-mEGFP-FLAG mutant exhibit robust disassembly of condensates formed during interphase, while overexpression of a phospho-dead TFG-T87A-mEGFP-FLAG mutant failed to dissolve entirely during mitosis, supporting the role of hyperphosphorylation in the regulation of TFG condensates (Fig. 3f). These data together strongly support that TFG is a phosphoprotein that is regulated by phosphorylation and dephosphorylation events across the cell cycle, much like it was reported for TANGO1 in the early secretory pathway[31].

**Hydrophobic 'sticker' residues control the anisotropic condensation mechanism of TFG**
We next set out to identify which regions within TFG were responsible for the formation of hollow condensates (Fig. 4a). TFG was reported to assemble into homooctamers via an amino-terminal PB1 domain[11], and

is predicted by AlphaFold[33] to have two additional amphipathic alpha helices (Supplementary Fig. 8a, b). However, the majority of TFG is intrinsically disordered (73%; Supplementary Fig. 8c) and harbors a Q-rich region in which every third amino acid is glutamine. We purified recombinant TFG fragments with a high degree of purity (Supplementary Fig. 9a–f, i) that represent the folded moiety (TFG-1–123), the initial segment of IDR (IDR1: TFG124–240; fragments within IDR1: 124–161 and 162–240), the Q-rich tract (IDR2: TFG241–359), as well as the short carboxy-terminal fragment (IDR3: TFG360–400), and tested for the ability of each fragment to form anisotropic condensates (Fig. 4b). While the folded moiety of TFG (TFG1–123) failed to form condensates, all fragments within the intrinsically disordered moiety of TFG formed anisotropic, hollow condensates. Although the average diameter of fragment condensates largely matched the diameter observed for full-length TFG condensates, significant differences were detected in the size distribution among fragments (Fig. 4c; full-length TFG-1–400 condensates: 341 nm ± 87 SD, $n = 175$; diameter range of TFG fragments: 290–400 nm). Next, we performed FRAP experiments on condensates formed by TFG fragments and compared them to condensates formed by full-length TFG (Supplementary Fig. 9g, h). While condensates formed by the carboxy-terminal moiety of TFG (Q-rich TFG241–359 and TFG360–400) approached mobile fractions of ~25%, both full-length and individual amino-terminal fragments of TFG exhibited reduced mobility, which matched the low mobility observed within condensates formed by full-length TFG in vivo. TFG interacts with Sec16 via its amino-terminus[34] and TFG condensates are coated by Sec16 in cells (Supplementary Fig. 4c), suggesting that the amino-terminus of TFG faces the cytosol while its carboxy-terminus points to the condensate lumen. Furthermore, such an orientation would be consistent with a role proposed for the carboxy-terminus of TFG in COPII carrier uncoating[13]. These data provide evidence that multiple regions within the intrinsically disordered moiety of TFG can support the formation of anisotropic condensates.

We next set out to test whether electrostatic or hydrophobic interactions contribute to the anisotropic condensation mechanism by screening full-length TFG against increasing volume fractions of crowding agent and salt (Fig. 4d and Supplementary Fig. 10). TFG more readily condensed in the presence of elevated salt concentrations, i.e., screening out electrostatic interactions, pointing to the involvement of hydrophobic residues in the formation of hollow condensates. Strikingly, within IDRs 1-3, TFG harbors hydrophobic residues that exceed the typical spacing observed in coiled-coil proteins, and which are present in all truncations capable of forming anisotropic condensates (Fig. 4e). This organization was highly reminiscent of the recently proposed 'stickers and spacers' model that describes the mechanistic basis for phase separation of intrinsically disordered proteins[35] and suggested why individual fragments of TFG were able to support the formation of anisotropic condensates. The differential spacing of hydrophobic residues within TFG fragments may further cause the differences in size observed across condensates formed by TFG fragments. Importantly, cells transfected with full-length TFG in which

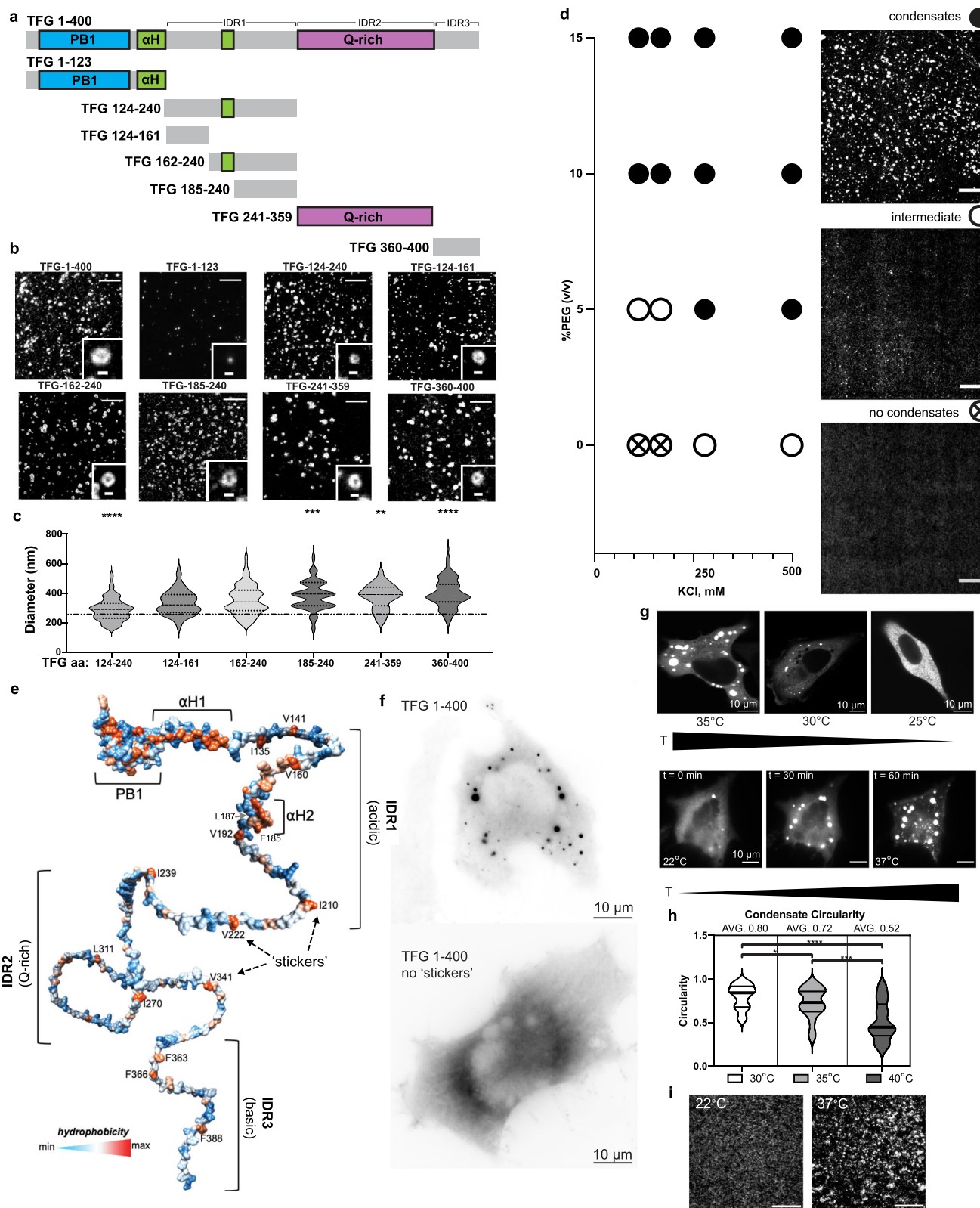

hydrophobic 'sticker' residues were deleted, condensates failed to form entirely in vivo, further highlighting the importance of these residues in the condensation mechanism (Fig. 4f). The relative strengths of 'sticker-sticker' interactions were shown to depend on their spacing across IDRs, as well as temperature[35,36]. Thus, we set out to probe whether TFG condensates obtained from overexpression would depend on temperature by shifting the cells from 37 °C to

defined temperatures (Fig. 4g). In cells shifted below 30 °C, TFG condensates disassembled and dispersed in the cytosol, while shifting the cells to 40 °C promoted condensate coalescence (Supplementary Fig. 11c). Strikingly, condensates at 30 °C were nearly spherical, with an increase in temperature resulting in a successive reduction in condensate sphericity concomitant with an anisotropic, sponge-like appearance (Fig. 4h). To mitigate the contribution of differential cell

**Fig. 4 | Hydrophobic 'sticker' residues control the anisotropic condensation mechanism of TFG. a** Schematic depicting domains present within TFG: PB1 domain, blue; alpha helices, green; Gln-rich region, purple; intrinsically disordered region (IDR), gray. Individual IDRs are defined as: IDR1 (net acidic), IDR2 (Gln-rich), IDR3 (net basic). Residues for truncated constructs are depicted to scale. **b** All truncated constructs except for residues 1-123 of TFG support the formation of lumen-containing (anisotropic) condensates. Representative micrographs are given. Overview scale bars 5 μm; insets 500 nm. **c** Quantification of the average peak-to-peak diameter of condensates formed by individual truncations of TFG. TFG; TFG 124–240: $n = 124$; TFG 124–161: $n = 79$; TFG 162–240: $n = 176$; TFG 241–359: $n = 93$; TFG 360–400: $n = 107$. The dashed line represents the average TFG 1–400 condensate diameter for reference. Two-tailed unpaired $t$-tests (TFG-124–240: $p < 0.0001$, TFG-185–240: $p = 0.0006$, TFG-241–359: $p = 0.0011$, TFG-360–400: $p < 0.0001$). **d** TFG phase diagram probing increasing concentrations of KCl vs PEG

8 kDa as a crowding agent. TFG-mEGFP-FLAG: 70 nM; HEPES/KOH pH 7.3; X mM KCl as indicated; X% ($v/v$) PEG 8 kDa as indicated. Scale bar 10 μm. **e** Schematic based on an AlphaFold prediction of TFG structure color-coded for hydrophobicity (red: max; blue: min). Domain annotations and the position of hydrophobic residues ('stickers') within IDRs are given. **f** Widefield deconvolution micrographs of HeLa cells transfected with TFG-mEGFP-FLAG and TFG-1–400-w/o 'stickers'-mEGFP-FLAG, respectively. Scale bar 10 μm. **g** Representative micrographs of HeLa cells transfected with TFG-mEGFP-FLAG shifted to the indicated temperatures (top) and time-lapse of a single cell shifted through indicated temperatures (bottom). Scale bar 10 μm. **h** Quantification of condensate circularity at specific temperatures. Two-tailed unpaired $t$-tests ($^{*}p = 0.0378$, $^{***}p = 0.0003$, $^{****}p < 0.0001$) **i** Micrographs of TFG-162-240-mEGFP-FLAG in vitro incubated at indicated temperatures. Representative of three experiments. Scale bar 5 μm. Source data are provided as a Source Data file.

stress responses to temperature shock, single cells were tracked through a drop in temperature to 22 °C and a gradual shift back to 37 °C. Starting from a largely dispersed distribution of TFG in the cells at 22 °C, condensates rapidly formed as the temperature increased to 37 °C (Fig. 4g). Furthermore, the effect is recapitulated in vitro, wherein isolated TFG below its saturation concentration can be incubated at 37 °C and found to condense into anisotropic condensates from an initially dispersed distribution (Fig. 4i). These data suggest that a sticker-driven condensation mechanism controls assembly of TFG into hollow condensates, with interaction strengths among sticker residues maximizing near physiological temperatures.

Consistent with the ability of TFG to condense, we observed that overexpression of TFG diminished the amount of endogenous ERES and resulted in disassembly and fragmentation of the Golgi ribbon (Supplementary Fig. 11a). This suggests that endogenous TFG is sequestered into TFG condensates formed by overexpression, which would mimic the phenotype of TFG silencing and result in the previously reported defects in the organization of the early secretory pathway[10,19]. Strikingly, if 'sticker' residues are deleted in full-length TFG, endogenous ERES and Golgi markers remain unaffected, which strongly supports the specificity of TFG interactions via 'sticker' residues. Additionally, when both wildtype and 'sticker'-lacking variants of TFG are co-overexpressed, TFG condensates diminish with increasing concentrations of 'sticker'-lacking TFG (Supplementary Fig. 11b), which further supports that homotypic TFG interactions depend on hydrophobic 'sticker' residues.

### Hollow TFG condensates are porous and select for macromolecules of defined sizes

We next set out to further characterize TFG condensates by scanning electron microscopy (SEM) employing the anisotropic condensate-forming fragment TFG-185–240. Condensates appeared to have a uniform surface, suggesting enclosure of an aqueous lumen by a network of TFG. We next performed transmission electron microscopy (TEM) to obtain high-resolution images of TFG condensates in the absence of fixatives or crowding agents (Fig. 5a). Strikingly, the TEM analysis revealed the presence of irregular, <10 nm 'pore-like' openings on the surface of condensates (Fig. 5b), which prompted the hypothesis that proteins and cofactors sufficiently small could access the lumen of hollow TFG condensates. Next, we set out to test the relationship between molecular size and ability to access the lumen of TFG condensates by incubating TFG-185–240 condensates with fluorescent dextran species of increasing sizes (Fig. 5c and Supplementary Fig. 13). Dextrans smaller than 250 kDa readily accessed the condensate lumen, while larger dextrans were excluded—a feature that was preserved in condensates formed by full-length TFG (Supplementary Fig. 13).

We also purified a TFG fragment in which hydrophobic 'stickers' were removed (TFG185–240-w/o 'stickers'). Upon crowding, highly dynamic, spherical condensates formed that lacked the anisotropic

distribution of TFG, and importantly, failed to exclude dextrans of higher molecular weights, instead co-condensing with all molecular species of dextran tested (Fig. 5c, d and Supplementary Fig. 13). Thus, the porous structure and selective permeability of TFG condensates depend on its hydrophobic 'sticker' residues. Residues 174–184 encode for an amphipathic, alpha helical segment. While this region was not required for the formation of hollow condensates and neither affected the average size of openings on the condensate surface nor its selectivity for dextrans, its presence significantly improved yields and reduced condensate hardening, pointing to a role in stabilizing lateral interactions among TFG molecules.

### TFG condensates putatively insulate COPII from COPI coats

A size cutoff of 250 kDa corresponds approximately to a globular protein diameter of <8 nm[37], which could explain why the ER–Golgi interface excludes ribosomes[3,38] that are ~20 nm in diameter. We therefore set out to test whether fluorescently labeled ribosomes could access the lumen of TFG condensates in vitro. Strikingly, ribosomes were efficiently excluded from the interior of TFG condensates (Fig. 6b). Next, we obtained thin sections from cells exposed to scrambled siRNA and from cells in which TFG is silenced, and subjected them to TEM. Strikingly, we found a drastic significant increase in the number of vesicles associated with the ER–Golgi interface that appeared to spread over a larger area than in control cells (Fig. 6c, d). Furthermore, Golgi membranes appeared to be severely disrupted in cells lacking TFG, paralleling data reported in *C. elegans* that showed disruption of Golgi membranes and misplacement of both ERGIC and anterograde carriers when TFG was silenced[10]. These data support that TFG could serve as a barrier at the ER–Golgi interface that insulates carriers and excludes ribosomes.

It was previously speculated that TFG could condense to form a 'molecular glue' that populates the ER–Golgi interface and mediates adhesion between ERES and the Golgi[14], while promoting the uncoating of anterograde carriers[13]. We find that TFG indeed forms condensates at physiological conditions via hydrophobic 'sticker' residues, and assembles into hollow, porous spheres that may act as molecular 'sieves'. At the ER–Golgi interface, membrane carriers are routed in opposing directions at high rates[1]. While anterograde COPII carriers require the coat protein complexes Sec23/Sec24 and Sec13/Sec31 (each a stable heterodimer < 250 kDa), retrograde COPI carriers require coatomer (a heptameric complex > 500 kDa) that is recruited en bloc to membranes[5,39,40]. Our in vitro data suggest that TFG would be capable of functioning as a size-selective barrier that excludes COPI coats and allows passage of COPII coats. We set out to test this directly by overexpressing TFG to induce micron-sized condensates that allow for a simple scoring of colocalization with coat proteins. Next, we co-transfected COPII proteins Sec23, Sec24, or Sec23 and Sec24 simultaneously to probe whether they could enter TFG condensates (Fig. 6a and Supplementary Fig. 13e). Strikingly, we observe

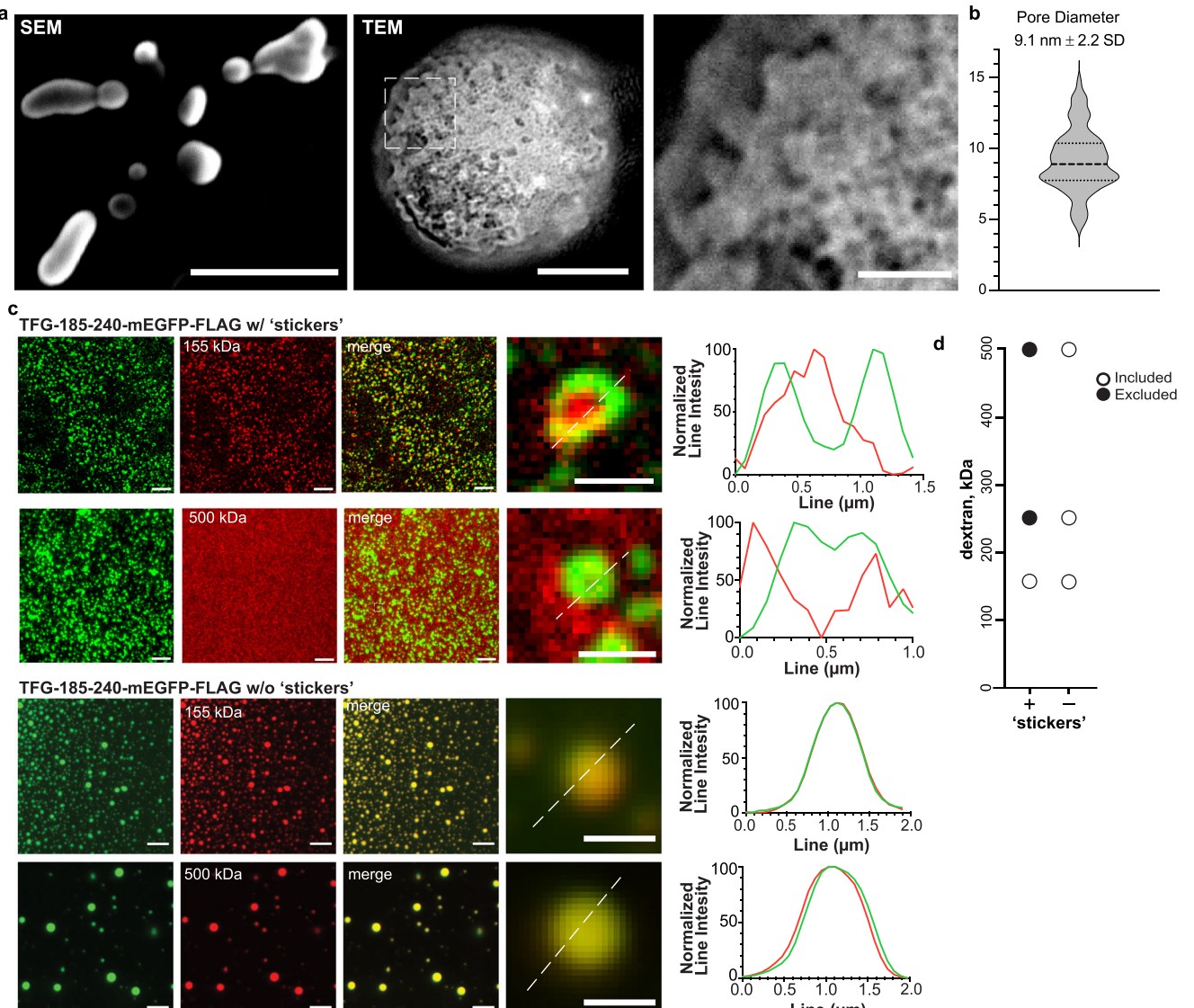

**Fig. 5 | Hollow TFG condensates are porous and select for macromolecules of defined sizes. a** Left: SEM of TFG-162–240-mEGFP-FLAG condensates. Scale bar 500 nm. Middle: transmission electron micrograph (TEM) of a single TFG-162–240-mEGFP-FLAG condensate (no crowding agents or negative staining; LUT inverted). Scale bar 100 nm. Right: Magnification of the porous surface of TFG condensates (boxed area in middle panel, inverted LUT). Scale bar 25 nm. **b** Quantification of average pore size of TFG condensates micrograph ($n = 79$). **c** Permeability of TFG-185–240-mEGFP-FLAG and TFG-185–240-w/o 'stickers' -mEGFP-FLAG condensates (induced with 20 % ($v/v$) PEG 8 kDa) for fluorescent dextran-TMR species of various sizes. Representative confocal micrographs are given for 155 kDa and 500 kDa dextran. Normalized line intensities for representative merged images are given with TFG in green and dextran species in red. Scale bar 5 μm, zoom in 1 μm. Representative of at least three experiments. **d** Graph depicts the molecular weight range of dextrans that can access the lumen of TFG condensates (included: detected in the lumen of TFG condensates, white-filled circles; excluded: sequestered from TFG condensates, back circles). Source data are provided as a Source Data file.

an efficient sequestration of COPII coat proteins in all cases. Although an interaction between Sec23 and TFG has been previously identified[10], this has not been reported for Sec24, although they both appear to be enriched to the same extent. This may be due to a specific interaction of TFG with Sec24, or may result from multivalent interactions between IDR portions of Sec24 and TFG[41]. Additionally, when co-transfecting Sec23 and Sec24 at the same time, resulting in the formation of Sec23/24 heterodimers[42], Sec23/24 could enter TFG condensates with lower efficiency, but appeared enriched at the boundary of TFG condensates (Supplementary Fig. 13e). Interestingly, the presence of fluorescent proteins on both recombinant Sec23 and Sec24 would increase the overall dimer mass from 204 kDa to 262 kDa, approaching the predicted cutoff of ~250 kDa for TFG condensates determined in vitro and thus limiting access to the condensate lumen.

Next, we probed whether COPI coats could partition into TFG condensates by treating cells with Brefeldin A (BFA), which strips the COPI coat protein complex coatomer off the Golgi membrane[43] and thus increases the cytosolic pool of COPI available to encounter TFG condensates. Strikingly, upon immunolabeling coatomer, coatomer was consistently excluded from the TFG condensate dense phase. While treatment with BFA raised the concentration of cytosolic coatomer, which emphasized its exclusion from TFG condensates, exclusion was also observed in the absence of BFA. Furthermore, inclusion of COPII in TFG condensates with a concomitant exclusion of COPI was robustly observed for various condensate diameters (Supplementary Fig. 13f), including those matching the size of endogenous TFG condensates (Fig. 2a). Altogether, these data support that TFG condensates are indeed able to segregate COPII from COPI coats as predicted from our

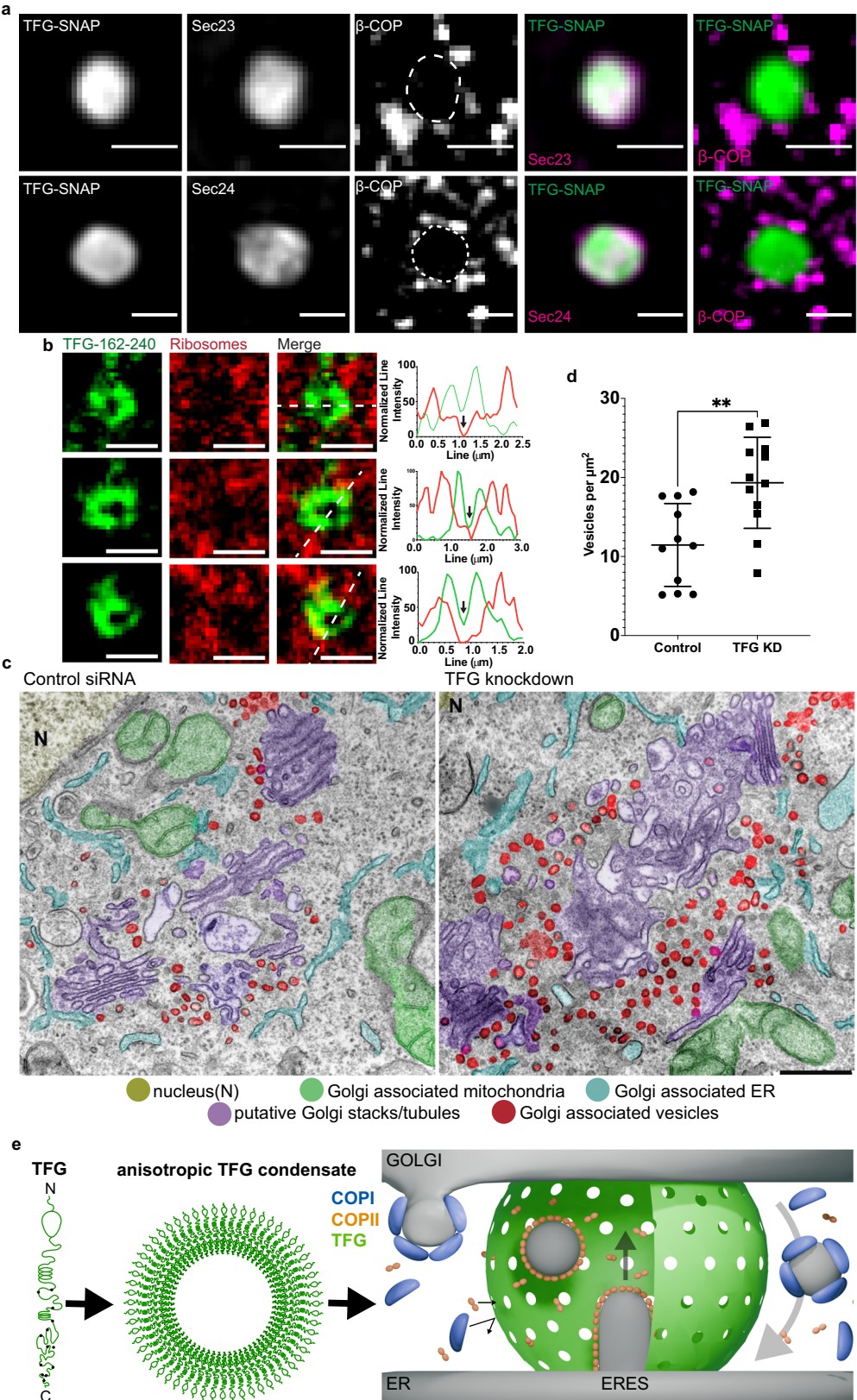

**a**

**b** TFG-162-240  Ribosomes  Merge

**d**

**c** Control siRNA          TFG knockdown

nucleus(N)   Golgi associated mitochondria   Golgi associated ER
putative Golgi stacks/tubules   Golgi associated vesicles

**e** TFG   anisotropic TFG condensate
COPI
COPII
TFG
GOLGI
ER   ERES

in vitro data. COPII coats can access the lumen of TFG condensates, putatively creating both a diffusion-limited space for anterograde transport towards the Golgi, while restricting COPI-mediated retrograde traffic to the periphery of TFG condensates (Fig. 6e).

Our findings address two long-standing questions regarding the anatomical and functional organization of the early secretory pathway.

First, TFG self-organizes to form anisotropic, 300-nm hollow condensates that serve as cytosolic extensions of ERES, thereby structuring and controlling the dimensions of the ER–Golgi interface and putatively contributing to membrane concavity at both ERES and *cis*-Golgi membranes, both representing anatomical features that are conserved across multiple taxa[2,3,10]. This further results in TFG creating

**Fig. 6 | TFG condensates insulate COPII from COPI coats. a** Representative micrograph of HeLa cells transfected with FLAG-TFG-SNAP: Snap-Cell 647 SiR, mEmerald-Sec23A (top), EYFP-Sec24C (bottom), and immunostained for β-COP, a component of coatomer. Scale bar 1 μm. The experimental setup was reproduced seven times. **b** Permeability of TFG-162–240-mEGFP-FLAG condensates (induced with 10% (v/v) PEG 8 kDa) for *E. coli* ribosomes labeled with NHS-SiR. Line scans depict normalized line intensity with TFG in green and ribosomes in red. Scale bar 1 μm. Representative of at least three separate experiments. **c** Electron micrographs of (left) control siRNA and nocodazole and (middle/right) TFG knockdown and nocodazole. Nucleus (N), yellow; Golgi-associated mitochondria, green; Golgi-associated ER, cyan; putative Golgi stacks/tubules, purple; Golgi-associated

vesicles, red. **d** Quantification of vesicles per μm² of multiple selected Golgi stack areas from micrographs depicted in c. p = 0.0025. See "Methods". **e** Model depicting the role of TFG at the ER–Golgi interface: TFG self-organizes into 300-nm condensates that are hollow. Their porous surface allows for COPII coat components (Sec23/24, Sec13/31, both complexes < 250 kDa) to access the lumen of TFG condensates, forming anterograde carriers. The retrograde carrier-forming COPI coat (550 kDa total) cannot access the lumen of TFG condensates. This may lead to a spatial segregation of anterograde from retrograde traffic, and the creation of a diffusion-limited space for bidirectional transport at the ER–Golgi interface. Source data are provided as a Source Data file.

a diffusion-limited space between the ER and Golgi and may contribute to the overall polarization of the early secretory pathway. Furthermore, TFG condensates are regulated by phosphorylation, resulting in their dispersal upon mitotic entry when they become hyperphosphorylated. Second, based on discontinuities present on the surface of hollow TFG condensates, bidirectional transport at the ER–Golgi interface may be spatially segregated to ensure an efficient exchange of cargo between the organelles. This would prevent an aberrant diffusion of anterograde carriers away from target Golgi membranes, while avoiding collisions between uncoated anterograde and retrograde carriers. TFG condensates may act as molecular 'sieves,' which would be supported by the exclusion of ribosomes from the ER–Golgi interface and the segregation of vesicle-forming machinery (with COPII coats localizing to the interface, and COPI coats segregated to its periphery), both of which we confirmed in vitro or in vivo for TFG condensates formed via overexpression. Once assembled in the condensate interior, the COPII-coated carriers formed from Sec23/Sec24 and Sec13/31 heterodimers would exceed the exclusion size of the TFG boundary, leading to their concentration at the ER–Golgi interface. The resulting coalescence of anterograde carriers within the lumen of hollow TFG condensates could contribute to the formation of the ER–Golgi intermediate compartment (ERGIC)[44], with energy-dependent docking and fusion machinery overcoming the TFG boundary to mediate content delivery to target membranes[45-47]. In agreement with these points, when TFG is depleted, the symmetry of the ER–Golgi interface is lost, resulting in an accumulation of vesicles outside the interface and a reduction of Golgi membranes concomitant with a reduced flux from the ER[10,19]. Notably, TFG condensates serving as extensions of ERES would be compatible with recently proposed 'tunnels' and/or tubules that may enable export of bulky cargo[17,48]. Importantly, our results are also compatible with recent reports that propose self-organization of transmembrane and peripheral ER–Golgi interface components[12,15,16,18]. Altogether, these findings lead to an emerging view in which the early secretory pathway is structured by protein collectives capable of dictating its morphology, connectivity, and performance.

## Methods

### Cell culture/transfection/labeling

HeLa cells (ATCC, CCL2) were grown at 37 °C at 5% CO₂ in Dulbecco's modified Eagle medium (Gibco 10566-016) supplemented with 10% FBS (Gibco A31604-01). Cells were seeded on glass-bottom Mattek dishes (P35GC-1.5-14-C) for live cell imaging or fixed with 4% PFA (Electron Microscopy Sciences, cat # 50980487) for 15 min. Fixed cells were then washed with PBS and permeabilized with permeabilization buffer (0.3% IGEPAL, 0.05% Triton-X 100, 0.1% BSA), then washed with wash buffer (0.05% IGEPAL, 0.05% Triton-X 100, 0.2% BSA), and blocked with blocking buffer (0.05% IGEPAL, 0.05% Triton-X 100, 5.0% normal goat serum) for 1 h at room temperature. As indicated, cells were labeled with anti-GM130 (BD Biosciences, cat # BDB610822), anti-SEC16 (Invitrogen, cat # PA552182), anti-TFG (Abcam, cat # 156866), anti-TANGO1 (Abcam, cat # 244506) for 1 h, washed, and labeled with goat-anti-mouse (Invitrogen, cat # A32727) and goat-anti-rabbit

(Invitrogen, cat # A32732) Alexa Fluor Plus–conjugated secondary antibodies for 1 h, subsequently washed and imaged. For ER–Golgi interface diameter measurements (Fig. 1g), *n* > 3 cells were utilized with multiple measurements derived from each.

For overexpression of TFG (Figs. 2 and 4), cells were transfected with TFG-mEGFP-FLAG at 1 μg/μL using Fugene (Promega, cat # PRE2311) and left to express for 24 h. 4-color micrographs (Fig. 1a) were obtained from HeLa cells co-transfected with mGFP-Sec16L at 3 μg/μL and FLAG-TFG-SNAP at 1 μg/μL and subsequently labeled with SNAP-Cell 647-SiR (NEB, S9102S) and immunostained with anti-GM130, anti-TANGO1, goat-anti-mouse Alexa Fluor Plus 405 (Invitrogen, cat # A48255), and goat-anti-rabbit Alexa Fluor Plus 555. Additional antibodies used in this study include: anti-GFP (Sigma, A11122) and anti-FLAG (Sigma, F1804). For COPI/II inclusion-exclusion experiments, cells were transfected with FLAG-TFG-SNAP at 0.5 μg and mEmerald-Sec23A, EYFP-Sec24C, or mEmerald-Sec23A + EYFP-Sec24C at 0.5 μg, totaling 1 μg. TFG was then labeled with Snap-Cell 647 SiR at 200 nM, and cells were immunostained with anti-β-COP (Abcam, AB2899). For BFA treatment, cells were incubated with 10 μM for 1 hr prior to fixation with PFA (Fig. 6a and Supplementary Fig. 13e).

For growth curve experiments, an siRNA transfection complex was prepared using TFG siRNA (L-016366-00-0005, Horizon Discovery Biosciences Limited). Oligonucleotides were diluted to 20 μM in ddH₂O, and 10 μL were mixed into 175 μL of Opti-MEM (for mock knockdown, siRNA was omitted). Next, 3 μL of oligofectamine (Cat# 12252-011) was mixed into 12 μL of Opti-MEM. Both solutions were incubated for 5–10 min. Solutions were combined, incubated for 20 min, and 800 μL of Opti-MEM was added for a total of 1 mL of transfection mixture. 500,000 cells were spun at 200×g, resuspended in 1 mL of transfection mixture, and added to target cells for 20 h in a 6-well plate. After incubation, cells were spun down at 200×g, and resuspended in 2 mL of Expi expression medium. An automatic cell counter (Countess II; Thermofisher) was employed to analyze cell counts at the indicated time points.

### Endogenous tagging of TFG in HeLa cells

Homozygous TFG::mClover-FLAG knock-in HeLa cells were generated by ExpressCells (ExpressCells, Inc., Philadelphia, PA, USA) via CRISPR/Cas9, and validated by PCR, Sanger sequencing, Western blotting, and fluorescence microscopy.

### Microscopy

Conventional (widefield/deconvolution, confocal) and super-resolution microscopy were performed on a Cytiva OMX SR microscope setup equipped with an Olympus PlanApo N 60X/1.42 oil objective. Multi-channel alignment was performed by the manufacturer using Tetra-Speck beads. Multicolor micrographs were aligned employing OMX-specific software packages (softWoRx 7.2.2). Images of immunolabeled cells (Supplementary Fig. 4a–c) were obtained using widefield microscopy. For time-lapse imaging of transfected HeLa cells (Supplementary Fig. 3a), images were acquired once every minute for 60 min using conventional widefield mode. Images of sponge-like TFG condensates

(Supplementary Fig. 4b) were obtained via 3D structured illumination microscopy (3D-SIM), and images were reconstructed employing OMX-specific software packages (softWoRx).

SEM images (Fig. 5a) were taken on an FEI Quanta 450 with a working distance of 10 mm, 5–10 kV, aperture 7. The sample was blotted on 0.1 μm filters, mounted on 12 mm Al stubs (Ted Pella, Redding, CA, cat# 16111) using double sticky carbon tabs. Samples were coated with 6 nm platinum using a Quora 150 V coater. TEM images (Fig. 5a and Supplementary Fig. 14a, b, c, f) were taken on an FEI Tecnai T12, 120 kV, with an AMT digital camera. The sample was blotted onto 100 mesh carbon-formvar coated grids (Electron Microscopy Sciences, Hatfield, PA, cat # FCF100H-Cu). For vesicle analysis in thin sections, cells were fixed in Karnovsky's fixative for 24 h and processed for monolayer embedding and sectioning. Grids were stained with uranyl acetate and lead nitrate prior to viewing.

Electron micrographs were false colored (Fig. 6c), and individual Golgi stacks were marked and measured for area and subsequently quantified (Fig. 6d) for putative COPII vesicles. The criteria for Golgi stack selection were visible boundaries such as mitochondria and, in particular, RER. Vesicles exiting the Golgi lumen were excluded, whereas vesicles obviously exiting the ER were included.

All in vitro confocal, 3D-SIM, and TIRF images were taken with the sample spotted on poly-D-lysine-coated glass bottom dishes and at RT unless otherwise indicated (Mattek, P35GC-1.5-14-C).

Diameter measurements (Figs. 1g, 2b, and 4c) were taken using line scan measurements between pixels with the highest brightness values. Homozygous HeLa TFG-mClover-FLAG cells were seeded on glass-bottom Mattek dishes (P35GC-1.5-14-C) for live cell imaging with live imaging solution (Invitrogen A1429IDJ) supplemented with 10 mM final concentration of glucose (Sigma G8644-100ML). Cells were labeled with Hoechst 33342 (Invitrogen, cat # H3570) at a concentration of 5 μg/mL for 5 min. Micrographs of labeled cells (Fig. 2a) were obtained using widefield microscopy.

STED images were produced from HeLa cells seeded on glass-bottom Mattek dishes (P35GC-1.5-14-C) and transfected with TFG-mEGFP-FLAG at 1 μg/μL using Fugene (Promega, cat # PRE2311) for 24 h. Cells were fixed with 4% PFA (Electron Microscopy Sciences, cat # 50980487) for 15 min and mounted with Prolong Diamond antifade mountant (Invitrogen cat # P36970). STED micrographs were obtained using an Inverted Leica DMi8 (UCSD Microscopy core) with 100x oil objective (excitation laser: 484 nm; depletion laser: 592 nm).

For FRAP analysis of TFG condensates in cells (Fig. 2), cells were transfected as described above, and individual TFG condensates from $n > 3$ cells were either fully or partially bleached using the FRAP module in the OMX SR platform combined with confocal mode at 37 °C. FRAP of in vitro TFG condensates from independent purifications was recorded at RT. Recovery was measured using the Time Series Analyzer V3 plugin in ImageJ[39], and statistical analysis and plots were generated using GraphPad Prism 10.

### Protein purification from Expi293 cells

Full-length and truncated TFG-encoding plasmids were transfected into Expi293F cells (ThermoFisher) at 1 μg/mL culture using Expifectamine 293 Reagent per manufacturer's instructions. Time course experiments were conducted to ensure optimal expression time for constructs, with expression times varying from 16 h to 72 h. Expi293F cells were incubated at 37 °C, 8% $CO_2$ on a shaker at 120 rpm. If indicated, cells were synchronized using a double thymidine block. Cells were pelleted at 800×g for 10 min and frozen in liquid nitrogen. Pellets between 15 mL and 60 mL of culture, depending on the construct, were resuspended in 10 mL of buffer 1 [50 mM HEPES/KOH, pH 7.3, 150 mM KCl, EDTA-free protease inhibitor cocktail tablet (Roche)]. Cells were then homogenized employing 20 passages through a 25 G needle. Lysate was centrifuged at 800×g for 30 min. Pelleted material was

discarded, and 5 mL buffer 2 [50 mM HEPES/KOH, pH 7.3, 1.2 M KCl, EDTA-free protease inhibitor tablet (Roche)] was added to the supernatant to adjust the salt concentration to 500 mM KCl final concentration. Salt-adjusted lysate was then centrifuged at 20,000×g for 15 min. Pelleted material was discarded, and the supernatant was transferred to a conical tube containing 2.5 mL blocked FLAG-affinity resin equilibrated with buffer 3 [50 mM HEPES/KOH, pH 7.3, 500 mM KCl, EDTA-free protease inhibitor tablet (Roche)] and rotated for 1 h at room temperature. The suspension was settled on a column that was equilibrated with 50 mL buffer 3. The resin was then washed with 100 mL buffer 3. Next, the column was drained, and 1 mL buffer 4 [buffer 3 plus 0.5 mM ATP, 0.5 mM $MgCl_2$ (Thermofisher)] was added to the resin and incubated for 10 min, followed by a wash of 50 mL buffer 3. If RNAse wash is indicated, 10 U of RNAse A (Thermofisher, Cat# 19101) was diluted into 1 mL total of buffer 3, added to the resin, incubated for 10 min, and followed by a wash of 50 mL of buffer 3. The proteins were eluted in buffer 5 [buffer 3 plus 200 μM FLAG peptide (Sigma-Aldrich)] for 1.25 h, obtaining 5 fractions total. Elutions were analyzed on 4–20% Bis-Tris gradient gels, stained with Coomassie, and analyzed on a LI-COR Odyssey infrared scanner utilizing LI-COR Image Studio 5.2. After elution, fractions were frozen in liquid nitrogen before further processing. Quantitative Western blot analysis was employed to determine the concentration of recombinant protein using a recombinant GFP standard (Roche, 11814524001) with a defined concentration of 1 mg/mL. Comparative Coomassie staining in conjunction with background-subtracted quantification of band intensity (LI-COR) of standards was employed to fit a gel-specific standard curve used to determine the concentration of a dilution series of candidate proteins analyzed on the same gel. SuperSep Phos-tag (50 μmol/l), 7.5% gels (Fuji Film, 192–18,001) were used to analyze the phosphorylation state of TFG-mEGFP-FLAG per manufacturer's instructions. Western blotting of Phos-tag gel and TFG-162-240-mEGFP-FLAG was performed using Thermo Fisher anti-TFG antibody (cat# A302-342A-T).

Concentration of samples was performed using a 50 kDa Amicon Ultra-0.5 Centrifugal Filter Unit (Millipore, UFC5050). The elution was thawed at 37 °C with occasional vortexing, and the sample was centrifuged at 21.1×g for 10 min before concentration. The column was washed in 200 μL steps with 2 mL total of 50 mM HEPES/KOH, pH 7.3, and 150 mM KCl, yielding a final 150 mM KCl in the sample. Next, the sample was concentrated further stepwise in the desalting column, crowding agents and reductive agents were added as indicated, and the sample was used immediately for experiments without further storage.

ColabFold v1.5.2-patch (AlphaFold2) was used for AlphaFold models of proteins.

### Probing size selectivity of TFG condensates

Purified TFG-mEGFP-FLAG (+truncations) in 50 mM HEPES/KOH, pH 7.3, and 150 mM KCl was pipetted onto standard glass-bottom MatTek dishes (P35GC-1.5-14-C) to form condensates (with 20% $v/v$ PEG 8 kDa as needed) and overlaid with either fluorescent dextrans or ribosomes. Fluorescent dextrans of various sizes at 70 kDa (Invitrogen D1819), 155 kDa (T1287-100MG, SIGMA-ALDRICH, INC.), 250 kDa (TMR-dex 250 kDa, Fina Biosolutions), and 500 kDa (52194-1 G, SIGMA-ALDRICH, INC.) suspended in 50 mM HEPES/KOH, pH 7.3, and 150 mM KCl. Dextran species were added at a final concentration of 1 μM and incubated for 10 min. Ribosomes (NEB, cat# P0763S) were labeled with NHS-SiR in approximately a 1:4 molar ratio, washed with 4 mL 50 mM HEPES/KOH, pH 7.3, 150 mM KCl, and 10 mM $MgCl_2$. Purified TFG-162-240-mEGFP-FLAG was pipetted onto a lysine-coated glass-bottom MatTek dish (MatTek, P35GC-0-10-C) to form condensates (with 10% $v/v$ PEG 8 kDa) and overlaid with equal volume fluorescently labeled ribosomes and incubated for 10 min. All images were taken in confocal mode using the OMX Flex SR microscope.

### In vitro phosphatase assay

Purified TFG-162-240-mEGFP-FLAG was incubated per manufacturer's instructions in 50 mM HEPES/KOH, pH 7.3, 150 mM KCl, supplemented with 2 mM $MnCl_2$ and 1 µL of lambda protein phosphatase (NEB, cat# P0753S) and incubated at 30 °C for 30 min. Recombinant protein was then pipetted onto a 35 mm poly-D-lysine coated MatTek dish to observe potential condensate formation.

### RUSH assay

HeLa cells (ATCC, CCL2) were seeded onto glass-bottom MatTek dishes (P35GC-1.5-14-C) for 24 h and subsequently transfected with TFG siRNA (L-016366-00-0005, Horizon Discovery Biosciences Limited) and mock siRNA (D-001810-01-05, Horizon Discovery Biosciences Limited). Forty-eight hours post knockdown, dishes were transfected with 1 µg of Str-KDEL_SBP-EGFP-ECadherin plasmid and left to express for an additional 24 h. Cells were then imaged on the OMX SR platform in confocal mode at 37 °C, and in HEPES-buffered live-cell imaging solution (Invitrogen, A14291DJ) supplemented with 5 mM glucose. To release RUSH cargo from the ER, biotin (B4501, SIGMA ALDRICH, INC.) was added to a final concentration of 100 µM, and cells were imaged at 5-min intervals for 60 min. Statistical significance was determined via two-tailed, unpaired $t$-tests: $^{***}p < 0.001$, $^{**}p < 0.01$.

### Plasmids

The sequences for plasmids used were sourced from UniProtKB/Swiss-Prot and included: TFG Q92734. The fragments and T87 phospho-mutants for constructs used were generated via commercial gene synthesis (gBLOCK; IDT) and contained the sequence for mEGFP and FLAG. gBLOCKs and the pSNAP$_f$ plasmid (NEB, N9183S) were both digested with NheI (NEB, R0189S), removing SNAP and inserting the commercially obtained gene via ligation with T4 ligase (Sigma, 10481220001) per manufacturer's instructions. TFG-185–240-w/o 'stickers'-mEGFP-FLAG refers to TFG-185–240- (ΔF185, ΔL187, Δ192, ΔI210, ΔV222, and ΔI239)-mEGFP-FLAG. TFG-360-400-mEGFP-FLAG was generated via PCR using primers FWD:5'-tatatagctagcATGA-GACCAGGTTTTACTTCACTTCCT-3' and REV:5'-GATTACAAGGATGAC GACGATAAGctcgaggttaat-3'. The amplification product was gel-purified using Qiaquick Gel Extraction Kit (Qiagen, 28706×4), digested with NheI and XhoI (NEB, R0146S), and inserted into a similarly digested pSNAP$_f$ plasmid using T4 ligase. The ligation mixture was transformed into Subcloning Efficiency DH5alpha competent cells (ThermoFisher, 18265017), plated onto 100 µg/mL ampicillin agar plates (Biomyx, LA-2100), and several single colonies were cultured in LB and 100 µg/mL ampicillin. Single colony derived cultures were subsequently mini-prepped using a Qiaprep Spin Miniprep Kit (Qiagen, 27104), and plasmid sequences were confirmed via sequencing by Azenta/GeneWiz. pmGFP-Sec16L was a gift from Benjamin Glick (Addgene plasmid # 15776; http://n2t.net/addgene:15776; RRID:Addgene_15776). Str-KDEL_SBP-EGFP-Ecadherin was a gift from Franck Perez (Addgene plasmid # 65286; http://n2t.net/addgene: 65286; RRID:Addgene_65286). mEmerald-Sec23A was a gift from Jennifer Lippincott-Schwartz (Addgene plasmid # 166893; http://n2t.net/ addgene:166893; RRID:Addgene_166893). pEYFP-Sec24C was a gift from David Stephens (Addgene plasmid # 66608; http://n2t.net/ addgene:66608; RRID:Addgene_66608).

### Condensate thermoresponsiveness assay in live cells

The 100,000 CCL2 HeLa cells were seeded onto 35 mm glass-bottom Mattek dishes and allowed to adhere to the plate overnight. After 24 h, cells were transfected with 1 µg TFG-mEGFP-FLAG using the transfection reagents (Fugene + Optimem) per manufacturer's instructions and subsequently incubated in DMEM + 10% FBS. 24 h post-expression of TFG-mEGFP-FLAG, Mattek dishes were removed from the incubator, the medium was exchanged for 1 mL live-cell imaging buffer pre-incubated to defined temperatures, incubated for 5 min, and subjected to microscopy on an OMX DeltaVision SR setup in confocal mode. The axial dimensions of condensates for a given temperature were quantified to calculate average condensate circularity (ratio of width divided by length) per temperature condition. Individual cells that were tracked through the increasing temperature shift were prepared as noted above and imaged starting at 22 °C in 5 min intervals as the temperature increased to 37 °C.

### Condensate thermo-responsiveness in vitro

Recombinant TFG-162-240-mEGFP-FLAG was concentrated below the saturation concentration in 50 mM HEPES/KOH, pH 7.3, 150 mM KCl, incubated at 22 °C, and pipetted onto a poly-D-lysine coated MatTek dish imaged in confocal mode using an OMX DeltaVision SR. Recombinant TFG-162-240-mEGFP-FLAG was simultaneously incubated at 37 °C and then imaged in confocal mode.

### Mass spectrometry

All protein purified for mass spectrometry follows the protein purification protocol listed above with the addition of 3% Triton-X-100 present in buffer 2 (to adjust Triton-X-100 concentration to 1%) and 1% Triton-X-100 in buffer 3. Bands corresponding to recombinant TFG were excised after SDS-PAGE, and subjected to tryptic digest and mass spectrometry by Majid Ghassemian at the UCSD Biomolecular and Proteomics Mass Spectrometry Facility (National Institutes of Health [NIH] shared instrumentation grant number S10 OD021724). PEAKS software was used to analyze raw mass spectrometry data. Phospho-sites were included if present in at least two of three trials for each condition, if the -10LgP > 20, and if the Ascore > 19.

### Reporting summary

Further information on research design is available in the Nature Portfolio Reporting Summary linked to this article.

## Data availability

The data generated in this study are provided in the Supplementary Information/Source Data file. Further information and requests for resources should be directed to and will be fulfilled by the lead contact, Andreas Ernst (aernst@ucsd.edu). Source data are provided with this paper.

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

## Acknowledgements

We thank Dr. Arshad Desai, Dr. Elizabeth Villa, Dr. Enfu Hui, Dr. Ishier Raote, Dr. Ünal Coskun, and Dr. Aleksander Rebane for discussions and feedback on the manuscript. We thank Majid Ghassemian (UCSD Biomolecular and Proteomics Mass Spectrometry Facility) for performing our mass spectrometry experiments and Melina Brunelli for helpful discussions. This work was supported by NIGMS grant R35GM142433 to A.M.E. and by the funds from the University of California, San Diego. UCSD microscopy core is supported by NINDS P30NS047101. Biomolecular and Proteomics Mass Spectrometry Facility is supported by S10 OD016234 (Synapt-HDX-MS) and S10 OD021724 (LUMOS Orbi-Trap). W.R.W., M.R., and S.R.C. were supported by the UCSD Pathways in Biological Sciences Training Program, NIGMS T32 GM133351.

## Author contributions

S.M.B., W.R.W., M.R., S.R.C., S.N.C., and K.S.M.N. analyzed data and performed experiments. I.R.N. performed electron microscopy. A.M.E. conceived of the study and designed the research. W.R.W., S.M.B., M.R., S.R.C., and A.M.E. wrote the manuscript.

## Competing interests

The authors declare no competing interests.

## Additional information

**Supplementary information** The online version contains
supplementary material available at

Andreas M. Ernst.

peer review file is available.

