## [Transparent Peer Review file · Nature Communications]

A Hollow TFG Condensate Spatially Compartmentalizes the Early Secretory Pathway

Corresponding Author: Dr Andreas Ernst

Version 0:

Reviewer comments:

Reviewer #1

(Remarks to the Author)

In this paper, Wegeng and colleagues suggest that the ER-Golgi interface is organized by a porous condensate composed of TFG, which acts by selectively directing the transport of secretory carriers and segregating them from retrograde carriers, thereby optimizing membrane flux between the two organelles. While the idea is innovative, the evidence shown in support of this model falls well short of being convincing. Ultimately, the conclusions reached are quite premature, based on the data shown.

Comments:

1. TFG has been shown previously to form condensates both in cells and in vitro. The new finding presented in this study is that TFG condensates may be hollow. It is unclear why previous work did not reveal this feature. However, one distinction is that this study actually analyzes TFG-mEGFP-FLAG (or truncated fusions), as opposed to unmodified TFG. The authors should remove the tag from TFG after purification and directly label it for analysis in vitro. This will demonstrate whether TFG forms hollow condensates, or if its fusion to mEGFP-FLAG results in this effect.
2. The highly cytoplasmic distribution of the TFG::mClover-FLAG fusion that the authors generate via CRISPR contrasts numerous studies published previously. This suggests that TFG-mClover-FLAG is likely non-functional, and interpreting data using the fusion is problematic. Consistent with this, only a heterozygous clone is described. Unless the authors can demonstrate the functionality of this fusion, the data included in the manuscript using it are impossible to interpret and should be removed.
3. The physiological relevance of overexpressed TFG in cells is entirely unclear. Characterization of the condensates formed under these conditions likely has minimal relevance to condensates that form when it is expressed at endogenous levels.
4. The authors propose that phosphorylation sites across TFG may lead to repulsion and inhibit condensate formation. This is an exciting idea, which the authors should pursue (as opposed to the description of physiologically irrelevant condensates that form when TFG is overexpressed). It is entirely unclear why the study of TFG phosphorylation is not pursued, as it would have a significant impact on our understanding of TFG condensate formation.
5. Based on Figure 3, it appears that the authors define 'sticker' residues as the following 13 amino acids: I135, V141, V160, V192, I210, V222, I239, I270, L311, V341, F363, F366, and F388. However, in the methods, L187 and F185 are also cited as 'sticker' residues. This makes it impossible to understand what TFG 'no stickers' actually is. Moreover, it is perhaps unsurprising that deletion of all of these residues impacts TFG distribution in cells. Why are substitutions not made, as opposed to deletion? Additionally, rather troubling is the fact that the distribution of this mutant varies between figure panels (ie., Figure 3f looks entirely different from Extended Data Figure 8a), making it further difficult to know how the mutations are affecting TFG distribution.
6. The relevance of studying TFG-162-240-mEGFP-FLAG condensates by EM is unclear. Why not use full length TFG for this study instead? And why switch between different TFG fragments (162-240 vs 185-240) to test dextran permeability with and without 'sticker' residues? The inconsistencies here raise concerns about the generality of the findings.
7. The authors suggest that the ER-Golgi interface excludes ribosomes due to the presence of TFG. Thus, if TFG is depleted, ribosomes should invade this interface. The authors should test this idea directly, as they have all of the

appropriate materials to do so.

8. Similarly, if TFG is required to exclude COPI carriers from the ER-Golgi interface, then its removal should result in COPI and COPII no longer being spatially segregated. The authors should test this idea to validate their conclusion that TFG condensates serve as a “molecular sieve”. Otherwise, this idea should be removed from the manuscript, based on a lack of direct evidence.

Reviewer #2

(Remarks to the Author)

In this revision, the authors focus on the novelty of TFG to form hollow condensates; they examine this ability through endogenous labeling of TFG, exogenous introduction of TFG constructs with certain segments deleted, and purified TFG protein produced from human cells. The authors find that the TFG condensates are similar in size to the spacing between the ER and Golgi. They suggest that the condensates act as a molecular sieve that sorts components via size. The revision has focused more on the origins of the hollow condensates. However, the authors' findings are still preliminary. As will be elaborated below, it is not clear how short IDR segments (some as short as ~40 residues) scaffold in the appropriate manner to form hollow condensates. The authors point towards hydrophobic residues as the “stickers” that mediate condensate assembly, but it is not clearly described with many resulting questions. We suggest that a modeler / simulator may be important for this study. This would help bridge the gap in understanding how hollow condensates form, especially when using 40-residue TFG constructs fused to eGFP and FLAG peptide. Furthermore, the results of some experiments (e.g., temperature-based one in cells) may be difficult to interpret without complementary *in vitro* experiments or awareness of other cellular responses (e.g. heat shock or cold-stress response to temperature changes).

Major Concerns:

1) It is striking that three different IDRs are all individually sufficient to form anisotropic, hollow condensates. It is also striking that very short IDR segments (granted that they are fused to a ~27 kDa GFP and FLAG peptide) seem capable of forming hollow condensates. Given that these proteins were purified from human cells and that all three IDRs contain “stickers” that presumably recruit full-length, endogenous TFG, there is concern that there might be some minimal carryover of full-length TFG and/or other factors that contribute to this condensate morphology. It feels like something is missing from the interpretation of these experimental results. Recommendations are:

- a) Add anti-TFG western blot for at least the full-length construct and any deletion constructs for which there are suitable antibodies. This would be valuable since the tagged protein could be distinguished from the endogenous protein on the basis of size. This would at least address whether the endogenous protein co-purifies with the full-length recombinant one. Alternatively, the authors could run mass spectrometry of the various samples to determine if there is full-length TFG present, as well as to assess the post-translational modification state (potentially important given data presented in Figure 2).
- b) Given the putative interactions between TFG and the ER/Golgi interfaces, is it possible that purified TFG is bound to other things, such as lipids / membrane components?
- c) The authors should consider expressing TFG in bacteria (particularly the small IDR segments) if they have access to those resources. This would eliminate the risk of carryover of endogenous TFG and/or other eukaryotic secretion machinery. Replicating their results with proteins purified from a heterologous system would greatly strengthen their argument that these IDRs are able to form hollow condensates individually/autonomously.

2) It is striking that TFG 124-161 forms hollow condensates. What is the origin of the multivalency here necessary for assembly? What effect does the GFP fusion have on this assembly? What are the protein concentrations used for Figure 3? With such a short sequence, I'd encourage the authors to consider modeling how this IDR self-assembles. A deeper characterization is required since the data presented do not explain the molecular origins of the condensate hollowness.

3) Regarding the origins of the condensate-forming ability from hydrophobic residues – there are many questions. The authors mention that the condensates fail to form with their no-stickers construct in cells, but only show one image of this in Figure 3f. Additionally, the authors use temperature to highlight how cells form TFG condensates with increasing temperature, but they don't monitor a single cell over that temperature range (as is possible via live-imaging their GFP constructs). The temperature-based cell studies could be problematic for interpretation given differential cell response to cold shock and heat shock. The distribution of a condensation-prone protein in cells moved to temperatures above or below 37°C is at least as much a function of time as it is of temperature difference. Transient transfection is also inherently variable, so the same cell(s) should be tracked across time during temperature treatment and multiple cells with different expression levels should be shown if possible. The authors could and should do similar experiments with purified TFG constructs as a function of temperature to better understand the molecular origins of condensate formation. Additionally, the microscopy of no-stickers construct of TFG 185-240 in Extended Data Fig 8 should be directly compared against the microscopy of TFG 185-240 under similar conditions (crowding agent, temperature, protein concentration). There are similar condensates across both images (including hollow ones in Extended Data Fig 8).

4) The authors propose phosphorylation could regulate disassembly and that might be the reason they don't see FRAP recovery. Can they probe specifically for phosphorylated-TFG or perform mass spectrometry? Related to this comment, it's striking to see the difference in condensate status with Figure 2k. It is difficult to see the hollow condensates (if present) in Figure 2k.

5) Related to the PTM question above: Publicly available data with the TFG antibody they use shows a clear doublet band, and it almost looks like their band might be a doublet in the Coomassie stain they show for tagged overexpression of the full-length protein. What is the correlation between PTM status and the ability of TFG to form condensates, and how does this

relate to the different constructs of TFG made?

Other Concerns:

1) The correct language would be saturation concentration (threshold for when the protein begins to phase separate) as opposed to critical concentration.

2) In their endogenously tagged line, are all copies of TFG modified? Does the hollow ~300-400 nm structure formation inherent to TFG still work with all molecules fused to mClover-FLAG, or is only a subpopulation labeled? They do say "heterozygous" clones but only show a blot for the added tag. They should show an anti-TFG blot. They should also address the likely influence of untagged, endogenous protein on the localization of their tagged protein. They should indicate in the text that the endogenously tagged line is HeLa, as they've done in the associated figure legend and methods.

3) There are significant deconvolution artifacts in Figure 3f. Deconvolution should be repeated with different parameters or raw images should be shown instead. The same goes for extended data figure 3 panel a.

Reviewer #4

(Remarks to the Author)

My view is that the paper does make an advance over previous work on the topic. In my opinion, the major potential conceptual advance is that the condensates contain pores that are proposed to allow COPII subunit entry but not COPI, and thereby segregate anterograde carriers from retrograde ones at the ER-Golgi interface. This is based on EM and dextran exclusion data. The data are thus rather indirect on this point, which remains quite speculative. This is especially true if one considers that the pores appear heterogenous in size, commented on by another reviewer. In my opinion, further data is needed to strengthen the point that TFG condensates act as such a filter. Imaging COPII and COPI alongside TFG condensates in cells for example could be done, or biochemical evidence showing segregation of COPII versus COPI complexes by the condensates would also be good. In the absence of such new data, the advance shown in this paper could be seen as incremental, at least in conceptual terms.

Version 1:

Reviewer comments:

Reviewer #1

(Remarks to the Author)

The revised manuscript is a substantial improvement over the initial submission. The authors have provided stronger evidence that TGF forms 'sponge-like' condensates in vitro, dependent on its phosphorylation state. The evidence in cells is not as compelling however, particularly as the authors continue to analyze condensates that form following over-expression of the protein. Nonetheless, the mapping of phosphosites throughout the cell cycle is a very welcome addition, although further analysis of the T87 residue seems warranted, given its presence specifically in mitotic cells, where condensates do not form. It would be relatively straightforward to express a T87E mutant to determine whether it fails to de-condense properly during mitosis.

By contrast, the data suggesting that the sponge-like condensates insulate secretory carriers (COPII) from retrograde carriers (COPI) at the ER/Golgi interface continues to remain weak, particularly given how prominently this idea is presented in the study. The thin section EM data are difficult to interpret without immunogold labeling of COPII vs. COPI, and no quantification of the imaging data shown is provided. Without more direct evidence, the conclusion that TFG acts as a molecular sieve to prevent COPI vesicles from invading the space used by COPII seems premature.

Minor comment - there is a typo in line 334 ('byd' should be 'by')

Reviewer #2

(Remarks to the Author)

Thank you to the authors for thoroughly addressing concerns raised by all the reviewers. It is an interesting story likely to be impactful for understanding the ER-golgi interface. The work presents a potential new functional role for condensates in selective transport. There are still a few issues with the manuscript, some of which that arose from the revised figures and additional experiments. The molecular origins of the hollow condensate morphology remain unresolved, but potential biological explanation for the function of these hollow condensates is presented with additional experiments and rationale.

Concerns:

Protein concentration for the in-vitro images shown in Figure 1 (specifically 1E) are not shown or explained in the text. This hinders interpretation of their figures, especially across the main and extended data figures where multiple constructs and concentrations are tested. Does the zoom-in of Figure 1H highlight hollowness of condensates?

Have you tried collecting imaging movies to ascertain how coalescence of these droplets contributes towards formation of anisotropic condensates? This would really help address several questions concerning the hollow condensate morphology.

Extended data Fig 2CD are useful, but these are different constructs (TFG 162-240-mEGFP-FLAG condensates).

The image in Figure 2D and comparison with Extended Data Fig 6 makes it hard to see whether this is anomalous cell or a cell with very large condensates, especially in comparison to the images shown in Extended Data Figure 4A.

The addition of the phosphorylation studies has improved the manuscript. With the insights here, it does make this reviewer want to know more about what specific aspects of phosphorylation (and/or other modifications) are critical to the hollow condensate morphology (especially as bacterially-expressed protein does not form hollow condensates). It appears that the degree of phosphorylation coupled with which sites are phosphorylated are important to controlling condensation properties of TFG since hyperphosphorylation appears to inhibit condensate assembly for this protein. It would be important to iron this out for this paper, but I understand the complexities in doing so. The authors do frequently suggest that it is outside the scope of the current work. Readers will want to know the molecular basis for this porous type of condensate (possibly tied to changing material properties over time – see comment above regarding imaging movies).

The ribosome data in Figure 6B initially is confusing as it appears that ribosomes are recruited to the TFG condensate, as opposed to the exclusion the authors mention in the text. Additional replicates/images are required for this.

They still have different cells in their temperature ramp-down live imaging experiment of figure 4 (4G). They did follow the same cell for the temperature ramp-up experiment, but it appears to be a sick cell, given odd nuclear vacuolar morphology. Are there movies that the authors can supply for these experiments? It is interesting to note that these experiments do not exhibit hollow condensate morphology, but perhaps these do if you are able to image over longer time.

Reviewer #4

(Remarks to the Author)

The authors have performed additional experiments to address the point I raised about showing COPII and COPI coat segregation. These indicate that COPII subunits can enter condensates made upon TFG over-expression whereas COPI appears excluded. This would support the model whereby TFG spatially segregates ER to Golgi intermediates. However, I have several concerns over the nature of the data. The first relates to point that this is only seen with TFG overexpression. Numerous studies have observed endogenous COPII and COPI adjacent to each other in very close proximity at ERES and ERGIC in unperturbed cells. This does not fit with such a dramatic segregation as is shown upon TFG over-expression (Fig 6a). Some of the other data in Fig 6 is also not convincing e.g. ribosomes do not appear to be excluded from the TFG condensates shown in Fig 6b, contrary to the author's conclusions. The EM data in Fig 6c showing more vesicles in the vicinity of the ER and Golgi could be due to many reasons other than loss of TFG condensates in this region. For example, these could well be Golgi-derived vesicles considering that the Golgi morphology looks quite disrupted upon loss of TFG.

The model shown in Fig 6d showing a relatively large condensate at the ER-Golgi interface needs to be supported by analysis of the endogenous TFG protein in the absence of over-expression. This would appear to be key data that is necessary to avoid the criticism that the observations reflect an artificial situation that does not recapitulate the true organisation of the ER-Golgi interface.

Version 2:

Reviewer comments:

Reviewer #1

(Remarks to the Author)

The authors have addressed some of the concerns raised during this most recent round of review. A nice experiment showing the importance of T87 phosphorylation during mitosis is now included, and the authors have toned down the idea that TGF acts as a molecular sieve, as this is not directly shown. Since demonstration of this latter concept would have been a major conceptual advance for the field (it has already been shown by Qiu et al. that TGF undergoes phase separation to facilitate COPII vesicle trafficking), there is now a concern regarding the significance of this study, but this point is best left to the editors to evaluate.

Reviewer #2

(Remarks to the Author)

I thank the authors for clarifying points and modifying figures in this revised manuscript. I have no additional concerns.

Reviewer #4

(Remarks to the Author)

The authors have addressed my points in their new revised version of the manuscript. I would be happy to support publication.

REVIEWER COMMENTS

We would like to thank all Reviewers for their interest in our work and their constructive feedback on the previous version of our manuscript, which triggered several new lines of experiments that we believe greatly strengthen our conclusions in this second revision of our manuscript.

Reviewer #1 (Remarks to the Author):

In this paper, Wegeng and colleagues suggest that the ER-Golgi interface is organized by a porous condensate composed of TFG, which acts by selectively directing the transport of secretory carriers and segregating them from retrograde carriers, thereby optimizing membrane flux between the two organelles. While the idea is innovative, the evidence shown in support of this model falls well short of being convincing. Ultimately, the conclusions reached are quite premature, based on the data shown.

Comments:

1. TFG has been shown previously to form condensates both in cells and in vitro. The new finding presented in this study is that TFG condensates may be hollow. It is unclear why previous work did not reveal this feature. However, one distinction is that this study actually analyzes TFG-mEGFP-FLAG (or truncated fusions), as opposed to unmodified TFG. The authors should remove the tag from TFG after purification and directly label it for analysis in vitro. This will demonstrate whether TFG forms hollow condensates, or if its fusion to mEGFP-FLAG results in this effect.

It was reported that TFG can be salted out of solution to form reversible aggregates (Johnson et al, EMBO J 2015), but we respectfully disagree that it was demonstrated that TFG phase-transitions to form condensates either *in vitro* or *in vivo*, which our presented data establishes. In a recent publication by Kasberg et. al. 2024 in MBOC (Supplemental Fig. 1B, provided below as Reviewer Fig. 2) and Kasberg et. al. in Nature Communications in 2023 (Fig. 2a, 29 seconds, provided below as Reviewer Fig. 1), TFG endogenously tagged with Halo at its amino terminus was shown to form hollow structures, although the anisotropic nature of these condensates was not the focus of that publication. Halo-TFG partially colocalized with endosomal markers in this study, but also showed several condensates free of endosomal markers, reflecting TFG's moonlighting function at recycling endosomes in addition to its role at the ER-Golgi interface (Verdaguer et al., 2022; Peotter et al., 2022). We demonstrate that the anisotropic nature of these condensates persists *in vitro*, which could have been missed due to the approach prior research employed. Recombinant TFG was obtained from bacterial extracts, lacking the native post-translational modifications that protein purified from human cells provides. Among such modifications, we show that TFG purified from human cells is phosphorylated at multiple sites (new Fig. 3c) even during interphase, which likely contributes to electrostatic interactions that in addition to hydrophobic 'stickers' drive the specific assembly of TFG into hollow spheres. We therefore believe that our *in vitro* data more closely reflects the structures observed *in vivo* based on endogenous tagging strategies, both in our own dataset as well as the recent literature.

While we did provide evidence in our previous manuscript that the anisotropy observed for TFG did not depend on the nature of the tag (Extended Data Fig. 4b) for overexpressed TFG *in vivo*, we agree with Reviewer 1's concerns that mEGFP-lacking, directly labeled TFG should be tested *in vitro*. We have purified full-length TFG-FLAG, which retains a minimal 8-aa epitope tag that we directly labeled with Alexa-647 NHS-ester (Extended Data Fig. 8). Notably, this protein assembles into the same structures as

observed with mEGFP-FLAG fusions, indicating that mEGFP does not induce anisotropy of TFG condensates.

2. The highly cytoplasmic distribution of the TFG::mClover-FLAG fusion that the authors generate via CRISPR contrasts numerous studies published previously. This suggests that TFG-mClover-FLAG is likely non-functional, and interpreting data using the fusion is problematic. Consistent with this, only a heterozygous clone is described. Unless the authors can demonstrate the functionality of this fusion, the data included in the manuscript using it are impossible to interpret and should be removed.

Previous data for TFG localization in cells was either based on immunofluorescence staining of endogenous TFG, with a loss of faint cytosolic signal likely owed to the use of detergents, or on overexpression of fusion protein, which we confirm results in distinct bright foci at ERES that are significantly brighter than the faint cytosolic pool. If TFG is indeed a biomolecular condensate, it would be expected to be observed as both dense and dilute phases inside cells, with the dilute phase representing the faint cytosolic pool, and the dense phase representing the hollow condensates.

Reviewer Response Fig 1. Both a cytosolic pool of TFG and distinct foci are detected for endogenously tagged Halo-TFG (Kasberg et. al. 2023; Fig. 2a).

Reviewer Response Fig 2. Hollow TFG spheres are detected in cells with endogenously tagged Halo-TFG (Kasberg et. al. 2024; Supplemental Fig. 1b).

However, we agree with Reviewer 1's concern that endogenous TFG might be rescuing a non-functional copy of TFG-mClover-FLAG. We have therefore obtained a homozygous TFG-mClover-FLAG clone, with which we have confirmed all key findings shown previously using the heterozygous cell line (Fig. 2a, b, 3a). We have provided an anti-TFG Western blot of the homozygous HeLa cells indicating that the homozygous clone lacks endogenous TFG (Extended Data Fig. 5). Thus, no endogenous TFG is present to rescue the hollow phenotype we see. Furthermore, we point again to Kasberg et. al. 2023 in Nature Communications (29 second inset) and Kasberg et. al. 2024 in MBOC, as the Halo-TFG knock-in cells seen in these publications mirror the phenotype we see closely. In these publications, Halo-TFG is also seen as a cytosolic distribution with small puncta present, several of which are hollow.

3. The physiological relevance of overexpressed TFG in cells is entirely unclear. Characterization of the condensates formed under these conditions likely has minimal relevance to condensates that form when it is expressed at endogenous levels.

We respectfully disagree that TFG condensates formed via overexpression are physiologically irrelevant. As previously shown with proteins such as GM130, FUS, hnRNPA1, and STING (Rebane et. al., FEBS Lett, 2019; Brangwynne et. al., Cell, 2019, Molliex et. al., Cell, 2015; Yu et. al. NCB, 2021), when the native condensates are obscured by the diffraction limit, overexpression can be used to induce larger condensates to aid in the investigation of their physical properties, e.g. via FRAP experiments, as we have also done with TFG. Whether one condensate is investigated or 10 coalesced, the condensate dense phase retains the same properties, and as such, physical parameters such as size exclusion or fluidity largely persist independent of condensate size. The use of studying overexpressed TFG condensates is supported by the fact that they retain anisotropy, representing a key feature of TFG condensates that is retained across various approaches in our dataset: 1) in supersaturated solutions (Fig. 1d), 2) for recombinant protein at physiological concentrations (Fig. 1h), 3) at endogenous levels in live cells (Fig. 2a, Extended Data Fig. 5b), 4) upon overexpression of recombinant proteins in cells (Fig. 2d, Extended Data Fig. 6a, b).

4. The authors propose that phosphorylation sites across TFG may lead to repulsion and inhibit condensate formation. This is an exciting idea, which the authors should pursue (as opposed to the description of physiologically irrelevant condensates that form when TFG is overexpressed). It is entirely unclear why the study of TFG phosphorylation is not pursued, as it would have a significant impact on our understanding of TFG condensate formation.

We thank reviewer 1 for this suggestion and have addressed this question with four new lines of experiments:

1) We have expanded our analysis based on TFG purified from synchronized expression cells to further probe their phosphorylation state during interphase and mitosis. While we had shown in the original manuscript that mitotic TFG fails to form condensates, we now show that mitotic TFG treated with lambda protein phosphatase (LPP) efficiently forms condensates that resemble interphase condensates (Fig. 3b). This effect was not due to crowding by LPP or altered buffer conditions, because interphase TFG below the saturation concentration did not assemble into condensates upon treatment with LPP (Extended Data Fig. 7a). These data directly demonstrate that TFG condensation can be toggled via phosphorylation in a minimal system *in vitro*.

2) We utilized a TFG construct in which all predicted phosphosites were mutated to negatively charged amino acids mimicking a phosphorylated state (i.e., serine residues were mutated to aspartic acid residues and threonine residues were mutated to glutamic acid). Supportively, we find that HeLa cells transfected

with this phosphomimetic TFG construct failed to form condensates, exhibiting instead a uniform cytosolic localization pattern (Fig. 3d and Extended Data Fig. 7b).

3) We analyzed mitotic and interphase TFG purified from synchronized cells via mass spectrometry. Surprisingly, phosphorylation of TFG was detected for both states with four phosphorylation sites persisting throughout both samples: D6, S50, S122, and S197, establishing that TFG is indeed a phosphoprotein, even if additional phosphorylation sites might be underestimated due to limitations in detection. Importantly, one phosphosite was identified with high confidence exclusively for mitotic TFG, T87, which indicates that TFG can be differentially phosphorylated across the cell cycle and may become hyperphosphorylated. Additionally, we find a significant detection of phosphorylation events in peptides containing the region AA204-AA215, which contains a cluster of repeated serine residues. This region is difficult to resolve via mass spectrometry due to the number of serine and threonine residues without neighboring lysine or arginine residues to allow for trypsin cleavage. While phosphorylation was robustly detected in this region, their location cannot be accurately assigned. However, we do suspect this region to be important for cell cycle mediated regulation of TFG condensation as we employed TFG-162-240-mEGFP-FLAG in our dephosphorylation assay, which harbors the serine-rich region (Fig. 3c).

4) We subjected recombinant TFG purified from mitotic and interphase cells to Phos-tag gel electrophoresis, followed by anti-TFG Western blotting. As depicted in Fig. 3d, several bands appear for both interphase and mitotic TFG that are absent in control gels/blots. Mitotic TFG exhibited a marked change in pattern, with disappearance of fast-running protein (i.e. low amounts of phosphorylation) and the appearance of bands corresponding to higher states of phosphorylation. This supports the interpretation that TFG is a phosphoprotein throughout the cell cycle but becomes hyperphosphorylated during mitosis. This parallels the regulation of other key proteins found in the ER-Golgi interface, e.g. TANGO1 was shown to become hyperphosphorylated to increase repulsion and induce disassembly of ERES during mitosis (Maeda, Dev. Cell, 2020).

5. Based on Fig. 3, it appears that the authors define ‘sticker’ residues as the following 13 amino acids: I135, V141, V160, V192, I210, V222, I239, I270, L311, V341, F363, F366, and F388. However, in the methods, L187 and F185 are also cited as ‘sticker’ residues. This makes it impossible to understand what TFG ‘no stickers’ actually is. Moreover, it is perhaps unsurprising that deletion of all of these residues impacts TFG distribution in cells. Why are substitutions not made, as opposed to deletion? Additionally, rather troubling is the fact that the distribution of this mutant varies between Fig. panels (ie., Fig. 3f looks entirely different from Extended Data Fig. 8a), making it further difficult to know how the mutations are affecting TFG distribution.

We are thankful to Reviewer 1 for bringing up these points and apologize for the confusion caused. We have added labels for F185 and L187 as sticker residues to our AlphaFold model, which were not visible due to the angle provided in former Fig. 3.

Regarding the cellular distribution of TFG-‘no stickers’-mEGFP-FLAG-transfected cells, we have added a gallery to support its localization pattern more clearly (Extended Data Fig. 12). The majority of cells exhibit a hazy cytosolic distribution, with a rare occurrence of cells showing small puncta (approx. 5% of cells) that are still distinct from wt TFG condensates (Extended Data Fig. 12; previously Extended Data Fig. 8).

To highlight the impact of ‘no stickers’ TFG on wt TFG, we performed competition experiments in which we titrated increasing amounts of wt TFG to TFG lacking sticker residues (Extended Data Fig. 11b). Both proteins can still oligomerize via the PB1 domain, meaning that increasing the amount of ‘no sticker’ TFG in heterooligomers should abolish condensation, if ‘sticker’ residues are indeed important for the condensation mechanism. While we previously showed such partial disassembly at 1:2 ratios of wt TFG to

TFG-no-stickers, we now include a 1:10 ratio of TFG:TFG-no-stickers that results in full dispersion of wt condensates. As described in our previous manuscript, the role of hydrophobic ‘sticker’ residues is further emphasized when monitoring the effect of overexpression of TFG, which results in disruption of the early secretory pathway, but only if ‘sticker’ residues are present (Extended Data Fig. 11a).

The choice of deleting residues rather than mutating them (e.g., to Ala) is to avoid a contribution of artificially introduced side chains (even if only a methyl group), that might impact condensation of TFG by altering the complex molecular grammar that contributes to phase separation of IDRs. We understand that this is unusual in regard to traditional “alanine screening” of proteins, however, we believe that in these cases it is warranted to reduce the parameter space when dealing with aggregation-prone, unstructured protein domains.

6. The relevance of studying TFG-162-240-mEGFP-FLAG condensates by EM is unclear. Why not use full length TFG for this study instead? And why switch between different TFG fragments (162-240 vs 185-240) to test dextran permeability with and without ‘sticker’ residues? The inconsistencies here raise concerns about the generality of the findings.

To this end, we have performed well over 150 purifications of TFG or TFG truncations in the past 3 years to acquire the data presented in our manuscript. While we were able to establish and reproduce purification of all variants, obtaining high yields of full-length TFG has been difficult. E.g., we do not store/flash-freeze concentrated protein at all and conduct all experiments immediately when it becomes available to avoid issues with stability of the protein. Despite these efforts, yields of full-length TFG are typically low, and tend to become unstable on the scale of hours, which is not uncommon for IDPs. However, based on the observation that TFG truncations can phenocopy the morphology of full-length TFG condensates (Fig. 4b) while retaining the size selectivity for model dextrans (Fig. 5c), we decided to focus on the truncation for EM experiments. Notably, between the constructs TFG-162-240-mEGFP-FLAG and TFG-185-240-mEGFP-FLAG, we saw a stability change which depended on the presence of the second amphipathic short alpha helix of TFG corresponding to residues 174-184. Many of the condensates formed from protein that lack alpha helix 2 hardened more rapidly while acquiring TEM images, although their selectivity for dextrans and porosity in EM was retained (Extended Data Fig. 14).

Thus, we respectfully disagree that the data is inconsistent, specifically since full-length and truncated TFG exhibit the same anisotropy, condensate size, and permeability for model macromolecules. Supportively, as outlined below, the ability of TFG to function as a ‘sieve’ for macromolecules is retained *in vivo* for COPI/II coats, and *in vitro* for ribosomes.

Focusing on truncated IDPs to study resulting protein condensates is not unprecedented in the LLPS field—it was demonstrated that for proteins such as FUS and DDX4, specific regions within the protein are sufficient to trigger condensation, establishing equivalence between truncated and full-length constructs that improve yields and thus facilitate probing higher concentrations of the target protein. Specifically, in Brangwynne et. al., Cell, 2018, full-length FUS was probed for condensation, but a region within FUS (FUS_N:residues 1-214) was created to generate the phase diagram seen in their study. Similarly, in Brangwynne et. al. 2017, Cell, DDX4 was truncated to DDX4_N, which was then used to characterize the phase separation of DDX4.

Nonetheless, we agree that the use of different constructs lengths for the dextran experiment is confusing, although we had offered the data for the sticker-containing TFG-185-240 construct in the Extended Data of the previous manuscript. We have now swapped the data showing dextran selectivity of TFG 162-240 for TFG 185-240 (Fig. 5c), matching the length of the corresponding ‘no sticker’ construct to improve clarity. For electron microscopy of condensates, we prefer to show TFG 162-240 in Fig. 5a as it does not

show extensive hardening compared to the TFG 185-240 construct when spotted on grids and viewed live in TEM (Extended Data Fig. 14).

7. The authors suggest that the ER-Golgi interface excludes ribosomes due to the presence of TFG. Thus, if TFG is depleted, ribosomes should invade this interface. The authors should test this idea directly, as they have all of the appropriate materials to do so.

We thank Reviewer 1 and agree that exclusion of ribosomes from TFG condensates, as predicted from our model, should be addressed. Exploring this question *in situ* at the ER-Golgi interface would require CLEM and ideally, FIB-milling-based cryo-ET. While this is an exciting area to explore in the future, our preliminary data suggests thus far that endogenously tagged TFG-mClover is too dim to serve as a guide for CLEM and FIB-milling, and thus creating cell lines with more photostable fusions such as Halo or Snap are beyond the scope of this study. As an alternative, we have addressed this question by incubating TFG condensates formed *in vitro* with Alexa647 NHS-ester labeled *E. coli* ribosomes. Ribosomes were efficiently excluded from the interior of the TFG condensates, mirroring the previously observed size exclusion of the dextran molecules, and thus supporting the interpretation that TFG is responsible for creating a ribosome-free space at the ER-Golgi interface (Fig. 6b).

8. Similarly, if TFG is required to exclude COPI carriers from the ER-Golgi interface, then its removal should result in COPI and COPII no longer being spatially segregated. The authors should test this idea to validate their conclusion that TFG condensates serve as a “molecular sieve”. Otherwise, this idea should be removed from the manuscript, based on a lack of direct evidence.

We thank Reviewer 1 for raising this point, and agree that TFG functioning as a molecular sieve for COPI and COPII coats required further experiments beyond those offered for model dextrans *in vitro*. We have addressed the validity of our model based on two new lines of experiments:

1) We addressed the permeability for COPII vs COPI coats directly in cells by inducing TFG condensates and probing them for inclusion of individual co-expressed COPII components or endogenous coatomer (Fig. 6a, Extended Data 13e). Specifically, TFG was overexpressed to induce micron-sized condensates that allow for a simple scoring of colocalization. Next, we co-transfected COPII proteins Sec23, Sec24, or Sec23 and Sec24 simultaneously to probe whether they could enter TFG condensates. Importantly, we observe inclusion of these coats in all cases. Although an interaction between Sec23 and TFG has been established (Witte et al, 2011) this has not been reported for Sec24, although they both appear to be enriched to the same extent. This may be due to the specific interaction of TFG with Sec24 or may result from multivalent interactions between IDR portions of Sec24 and TFG (Stancheva et al, 2020). Strikingly, when probing for inclusion of the COPI coat protein complex coatomer, we observed a consistent exclusion from the condensate dense phase. Since most endogenous coatomer resides at the Golgi, we immunolabeled endogenous coatomer after raising cytosolic levels via brefeldin A, which was not required to observe the exclusion per se but emphasized the effect. Additionally, when co-transfecting Sec23 and Sec24 at the same time, resulting in the formation of Sec23/24 heterodimers (Lederkremer et al., 2001), Sec23/24 could enter TFG condensates with lower efficiency, but appeared enriched at the boundary of TFG condensates. Interestingly, the presence of fluorescent proteins on both recombinant Sec23 and Sec24 would increase the overall dimer mass from 204 kDa to 262 kDa, approaching the predicted cutoff of ~250 kDa for TFG condensates determined *in vitro* and thus limiting access to the condensate lumen. Altogether, these data support that TFG condensates are indeed able to segregate COPII from COPI coats as predicted from our *in vitro* data. COPII coats can access the lumen of TFG condensates, thus creating both a diffusion-limited space for anterograde transport towards the Golgi, while restricting COPI-mediated retrograde traffic to the periphery of TFG condensates.

2) We compared the ultrastructure of control and TFG knockdown cells employing thin-section TEM. We robustly detect a significant increase in both the number of vesicles surrounding the ER-Golgi interface, as well as the area these vesicles occupy, both greatly supporting that TFG functions as a barrier to contain anterograde carriers (Fig. 6c, Extended Data Fig. 13e). Furthermore, the increase in the fraction of vesicles coincided with a disruption of Golgi membranes, paralleling a previous report in which TFG knockdown resulted in the mislocalization of anterograde carriers, ERGIC, and disruption of Golgi membranes in *C. elegans* (Witte et al., NCB, 2011).

Reviewer #2 (Remarks to the Author):

In this revision, the authors focus on the novelty of TFG to form hollow condensates; they examine this ability through endogenous labeling of TFG, exogenous introduction of TFG constructs with certain segments deleted, and purified TFG protein produced from human cells. The authors find that the TFG condensates are similar in size to the spacing between the ER and Golgi. They suggest that the condensates act as a molecular sieve that sorts components via size. The revision has focused more on the origins of the hollow condensates. However, the authors' findings are still preliminary. As will be elaborated below, it is not clear how short IDR segments (some as short as ~40 residues) scaffold in the appropriate manner to form hollow condensates. The authors point towards hydrophobic residues as the “stickers” that mediate condensate assembly, but it is not clearly described with many resulting questions. We suggest that a modeler / simulator may be important for this study. This would help bridge the gap in understanding how hollow condensates form, especially when using 40-residue TFG constructs fused to eGFP and FLAG peptide. Furthermore, the results of some experiments (e.g., temperature-based one in cells) may be difficult to interpret without complementary *in vitro* experiments or awareness of other cellular responses (e.g. heat shock or cold-stress response to temperature changes).

Major Concerns:

1) It is striking that three different IDRs are all individually sufficient to form anisotropic, hollow condensates. It is also striking that very short IDR segments (granted that they are fused to a ~27 kDa GFP and FLAG peptide) seem capable of forming hollow condensates. Given that these proteins were purified from human cells and that all three IDRs contain “stickers” that presumably recruit full-length, endogenous TFG, there is concern that there might be some minimal carryover of full-length TFG and/or other factors that contribute to this condensate morphology. It feels like something is missing from the interpretation of these experimental results. Recommendations are: a) Add anti-TFG western blot for at least the full-length construct and any deletion constructs for which there are suitable antibodies. This would be valuable since the tagged protein could be distinguished from the endogenous protein on the basis of size. This would at least address whether the endogenous protein co-purifies with the full-length recombinant one. Alternatively, the authors could run mass spectrometry of the various samples to determine if there is full-length TFG present, as well as to assess the post-translational modification state (potentially important given data presented in Fig. 2).

We thank Reviewer 2 for raising these fair points. Although our Coomassie gels do not indicate a significant presence of endogenous TFG or other protein contaminants for full-length and truncated TFG (Extended Data Fig. 9), we have performed Western blots employing anti-TFG antibody directed against residues 175-225, which enable analysis of both full-length TFG and the hollow condensate-forming truncation TFG 162-240 (Extended Data Fig. 2b, 9i). For full-length protein, minor signal is indeed detected at the size expected for endogenous protein, although a similar size product would be obtained if recombinant TFG-mEGFP-FLAG is degraded and lacks the mEGFP-FLAG moiety. Supportively, our anti-TFG Western blot against purified truncation TFG-162-240-mEGFP-FLAG does not show any signal at the size expected for endogenous TFG, suggesting that some, if not all putative endogenous TFG signal detected in full-length recombinant TFG could stem from degradation of the recombinant rather than co-purification of the endogenous protein. The latter data therefore strongly supports that anisotropic condensates formed by truncation constructs are not driven by copurified endogenous TFG. As outlined in response to Reviewer 1 point 1, we can exclude a contribution of the position and nature of the tag to the anisotropic condensate structure.

b) Given the putative interactions between TFG and the ER/Golgi interfaces, is it possible that purified TFG is bound to other things, such as lipids / membrane components?

The reviewer raised an intriguing point, but we have no evidence for contamination of lipids or membrane components in our purifications. Notably, the structures we saw were permeable to large molecules like fluorescent dextrans, which are membrane impermeable and thus supports that the recombinant protein lacks lipid contaminants at least to the extent at which a bilayer would template each condensate. Furthermore, any potential stabilizing effect from lipid contaminants contributing to our anisotropic phenotype would be unlikely due to the altered properties of our ‘no stickers’ constructs, which exhibit a loss of both anisotropy and selective permeability. Additionally, there is no indication based on our Coomassie gels that TFG variants attract a significant proportion of protein contaminants. Finally, Extended Data Fig. 2c of our original manuscript excluded that nucleic acids contribute to formation of hollow condensates.

c) The authors should consider expressing TFG in bacteria (particularly the small IDR segments) if they have access to those resources. This would eliminate the risk of carryover of endogenous TFG and/or other eukaryotic secretion machinery. Replicating their results with proteins purified from a heterologous system would greatly strengthen their argument that these IDRs are able to form hollow condensates individually/autonomously.

We thank the reviewer for this suggestion, however, when purified from bacteria, TFG was *not* able to form anisotropic condensates at physiological conditions. Instead, TFG purified from *E. coli* only exhibited a state reminiscent of condensates when it was salted out of solution using KOAc, forming amorphous aggregates that were reversible upon dilution with plain water (Johnson et. al. EMBO J, 2015). The bacteria-derived protein indeed does not reflect the behavior of the human cell-derived protein, which assembles at physiological conditions, resists washing and dilution once assembled, and yet can still disperse when hydrophobic stickers are targeted or when 1,6-hexanediol is added. Additionally, the new data included in this revision strongly points to the presence and importance of phosphorylation sites on TFG that are also present in interphase (Fig. 3c). These and other post-translational modifications that would be missing in bacterial protein but could affect the condensation mechanism (such as O-GlcNacylation and Arg methylation), make it likely that our observations are due to the fact that our recombinant protein stems from human cells, and thus reflects the endogenous PTM profile of human TFG. Understanding the exact contribution of hydrophobic and electrostatic interactions between TFG molecules that give rise to the unusual TFG condensate structure, and understanding how these are altered during mitosis, will be the focus of several new projects that we believe are not within a reasonable scope of this dataset.

2) It is striking that TFG 124-161 forms hollow condensates. What is the origin of the multivalency here necessary for assembly?

The multivalency may be encoded by the presence of three hydrophobic ‘sticker’ residues, which, together with electrostatic interactions encoded in the ‘spacer’ regions, might introduce valency to TFG monomers. Beyond this, ‘stickers’ might represent nodes that give rise to percolating networks, which would indeed be interesting to mathematically model in future studies, but we believe are beyond the scope of this manuscript. Furthermore, when simulating multiple TFG fragments using AlphaFold3, local folding patterns are detected that are not predicted for monomers, although we do not know to which extent these features are valid.

What effect does the GFP fusion have on this assembly?

As outlined for Reviewer 1 point 1, who shared the concern regarding a contribution of TFG, we believe this can be excluded. GFP-lacking, directly labeled recombinant TFG assembles into anisotropic condensates, and we had previously provided evidence that anisotropic condensates form when TFG is overexpressed in cells with a variety of tags (including epitope tags; Extended Data Fig. 4b).

What are the protein concentrations used for Fig. 3?

Protein concentrations for truncation variants of TFG are estimated at 50 nM per fusion protein as based on quantitative Western blotting, albeit with an error of approx. 20%. We thank Reviewer 2 for raising this point, and believe they are suspecting altered saturation concentrations of a given truncation (i.e., longer IDR fragments putatively have lower saturation concentrations than shorter fragments). While this might be the case, we would like to point out that anisotropic condensates depicted in Fig. 3 were obtained from incubations with the crowding agent 8 kDa PEG at 20% (v/v), meaning that while saturation concentrations could have differed among constructs, the fact that anisotropic condensates could be detected at all for all IDR truncations rather points to a common feature such as the aforementioned hydrophobic ‘sticker’ residues. This is supported by the fact that removal of such stickers for fragments in vitro abolishes the anisotropic distribution of protein in the dense phase (Extended Data Fig. 12c).

With such a short sequence, I’d encourage the authors to consider modeling how this IDR self-assembles. A deeper characterization is required since the data presented do not explain the molecular origins of the condensate hollowness.

We thank Reviewer 2 for suggesting exploring computational approaches such as modeling and simulation as additional evidence for the ability of short IDR segments to assemble into anisotropic structures. While we agree this is an interesting avenue to explore and are seeking collaborations with labs with expertise in atomistic molecular dynamic simulations, we believe that this warrants a full project rather than an ‘add-on’ to our manuscript and is thus beyond the scope of this publication. Furthermore, we are concerned that simulations of proteins undergoing LLPS might have limitations that do not always reflect real world experimental data, as suggested in a recent publication (Feito et. al., bioRxiv, 2024).

3) Regarding the origins of the condensate-forming ability from hydrophobic residues – there are many questions. The authors mention that the condensates fail to form with their no-stickers construct in cells, but only show one image of this in Fig. 3f. Additionally, the authors use temperature to highlight how cells form TFG condensates with increasing temperature, but they don’t monitor a single cell over that temperature range (as is possible via live-imaging their GFP constructs). The temperature-based cell studies could be problematic for interpretation given differential cell response to cold shock and heat shock. The distribution of a condensation-prone protein in cells moved to temperatures above or below 37°C is at least as much a function of time as it is of temperature difference. Transient transfection is also inherently variable, so the same cell(s) should be tracked across time during temperature treatment and multiple cells with different expression levels should be shown if possible. The authors could and should do similar experiments with purified TFG constructs as a function of temperature to better understand the molecular origins of condensate formation. Additionally, the microscopy of no-stickers construct of TFG 185-240 in Extended Data Fig 8 should be directly compared against the microscopy of TFG 185-240 under similar conditions (crowding agent, temperature, protein concentration). There are similar condensates across both images (including hollow ones in Extended Data Fig 8).

We thank Reviewer 2 for the constructive criticism, and fully agree that the points raised warranted additional experiments, which we have included in the revised manuscript:

- 1) We have added a gallery of HeLa cells transfected with full-length TFG-no-stickers-mEGFP-FLAG to our manuscript to support the statement about the phenotype (Extended Data Fig. 12a). Approximately 95% of cells show a cytosolic distribution of TFG-no-stickers-mEGFP-FLAG, and less than 5% exhibit small punctae which do not grow.
- 2) We recognize the differential effect that temperature shock of cells could have on condensate morphology due to stress responses. We have now tracked single cells through temperature changes to see the effect on condensate morphology (Fig. 4g). The data mirrors the original dataset such that cells exhibiting a dispersed distribution of TFG begin to form condensates that grow as the temperature approaches 37°C. We observe that > 90 % of tracked cells induce condensates as a function of temperature increase, which is in harsh contrast to cells simply overexpressing TFG, for which capturing phase transitions is rare.
- 3) Condensates readily formed when recombinant TFG-162-240-mEGFP-FLAG, below its saturation concentration, is shifted from 22°C to 37°C *in vitro* (Fig. 4i).
- 4) The conditions for TFG-185-240-no-stickers in Extended Data Fig. 12 and the conditions of TFG-185-240 in Fig. 4c are similar in terms of starting concentrations, crowding, and temperature. When dextrans are added, the sticker-lacking variant forms isotropic condensates that co-condense with the dextran, while > 75 % condensates formed by the sticker-containing counterpart were *anisotropic*. When sticker-lacking TFG 185-240 is sampled in absence of the dextrans, condensates again form that are predominantly isotropic (>85%), although occasionally, hollow condensates are indeed detected. Although the ‘stickers’ appear to play a major role in driving anisotropy of TFG condensates, it cannot be excluded that other features encoded in the ‘linker’ portions may contribute as well.

In future studies, we plan a deeper characterization of the exact contribution of ‘stickers and spacers’ to condensate morphology and permeability to increase our understanding of the molecular condensation mechanism. We believe this to be beyond the scope of this manuscript, which establishes

- i) condensation of TFG at physiological conditions
- ii) identification of the anisotropic nature of TFG condensates
- iii) regulation of TFG condensates via phosphorylation
- iv) the contribution of stickers to condensation mechanism of TFG
- v) permeability of TFG condensates, functioning as molecular ‘sieves’
- vi) an explanation how hollow TFG condensates can contribute to the morphology and function of the ER-Golgi interface by spatially segregating bidirectional traffic while forming a diffusion-limited space for anterograde transport.

4) The authors propose phosphorylation could regulate disassembly and that might be the reason they don’t see FRAP recovery. Can they probe specifically for phosphorylated-TFG or perform mass spectrometry?

We thank Reviewer 2 for the suggestion and agree that rather than suggesting phosphorylation could play a role in regulating TFG condensation, it should be directly experimentally addressed. As stated in our response to Reviewer 1 point 4 in detail, we have addressed this by adding four new lines of experimentation. In brief:

- 1) Dephosphorylation of mitotic TFG triggers its condensation *in vitro* (Fig. 3b; Extended Data Fig. 7a).
- 2) Phosphomimetic full-length TFG fails to form condensates upon overexpression in cells (Fig. 3e, Extended Data Fig. 7b).

3) Mass spectrometry analysis identifies TFG as a phosphoprotein with multiple phosphorylation sites, acquiring unique additional sites during mitosis (Fig. 3c).

4) Phos-tag gels combined with Western blotting confirms interphase TFG as a phosphoprotein, which becomes hyperphosphorylated during mitosis (Fig. 3d).

Related to this comment, it's striking to see the difference in condensate status with Fig. 2k. It is difficult to see the hollow condensates (if present) in Fig. 2k.

It is indeed striking that TFG purified from mitotic cells reproducibly fails to form condensates, which mirrors the dispersion of TFG foci observed in cells harboring endogenously tagged TFG-mClover (Fig. 3a). We thank Reviewer 2 for pointing out that hollow condensates are difficult to see in former Fig. 2k and have added a magnified inset to new Fig 3b to improve clarity.

5) Related to the PTM question above: Publicly available data with the TFG antibody they use shows a clear doublet band, and it almost looks like their band might be a doublet in the Coomassie stain they show for tagged overexpression of the full-length protein.

We agree with Reviewer 2 that a faint band may be migrating slightly slower than the main pool of eluted protein (current Extended Data Fig. 2a) is visible within Coomassie-stained elutions of recombinant TFG. Although this represents a minor fraction, it cannot be excluded that this represents a molecular species potentially harboring a different phosphorylation threshold. It is possible that these species are more emphasized in different SDS-PAGE conditions that may have been used for Western blotting in the publicly available data. However, in our hands, anti-TFG Western blots for our TFG-mEGFP-FLAG purifications do not show a significant fraction of this slower-migrating additional band (Extended Data Fig. 2b).

What is the correlation between PTM status and the ability of TFG to form condensates, and how does this relate to the different constructs of TFG made?

Based on the phosphorylation sites we are now able to identify via mass spectrometry, although likely representing an underestimate of all phosphosites in TFG, aa 1-123 contain 4 of the confidently assigned phosphosites, which may contribute to homooligomerization among PB1 domains. Notably, T87 is uniquely detected as a phosphosite for mitotic TFG, which is located within the PB1 domain as well. Thus, T87 may be regulatory and lead to dispersal of PB1 oligomers upon entry into mitosis. While we have shown that the folded moiety of TFG can neither form anisotropic condensates by itself, nor is it required to observe condensation of IDR fragments (Fig. 4b), it is possible that homooligomerization via PB1 *in vivo* reduces the saturation concentration significantly by aligning the IDRs of eight TFG molecules (Johnson et al. 2015). Furthermore, the poly-serine residue-containing cluster of amino acids between aa 204-215, whose phosphorylation could not be accurately localized or quantified using mass spectrometry, are of high interest for a candidate region toggling dispersion via repulsion among the IDR moieties. This region is retained in the construct used for the dephosphorylation assay (TFG-162-240-mEGFP-FLAG), and interestingly, corresponds to a region highly conserved among different taxa. We are dissecting the contribution of each detected phosphosite in TFG to its ability to form condensates in an ongoing project, which we believe is beyond the scope of this manuscript.

1) The correct language would be saturation concentration (threshold for when the protein begins to phase separate) as opposed to critical concentration.

We thank the reviewer for pointing out this error in our terminology and have edited the manuscript accordingly.

2) In their endogenously tagged line, are all copies of TFG modified? Does the hollow ~300-400 nm structure formation inherent to TFG still work with all molecules fused to mClover-FLAG, or is only a subpopulation labeled? They do say “heterozygous” clones but only show a blot for the added tag. They should show an anti-TFG blot. They should also address the likely influence of untagged, endogenous protein on the localization of their tagged protein. They should indicate in the text that the endogenously tagged line is HeLa, as they’ve done in the associated Fig. legend and methods.

As pointed out by Reviewer 1 (point 2) as well, we agree that the use of a heterozygous clone raises several concerns, and therefore have repeated all experiments with a homozygous TFG-mClover-FLAG HeLa clone. We find the same distribution of dilute and dense phases inside cells as for the heterozygous clone and can confirm that characteristics regarding the size and dynamics of condensates remain the same (new Fig. 2a, b, c, Extended Data Fig. 4b). As suggested, we have added an anti-TFG Western blot for both control HeLa and homozygous TFG-mClover-FLAG knockin HeLa cells (Extended Data Fig. 5), confirming that endogenous TFG is fully edited in the homozygous clone.

We thank Reviewer 2 for pointing out that we omitted to mention the cell type in the main text, which has been corrected in the revised manuscript.

3) There are significant deconvolution artifacts in Fig. 3f. Deconvolution should be repeated with different parameters or raw images should be shown instead. The same goes for extended data Fig. 3 panel a.

We thank Reviewer 2 for pointing out the deconvolution artifacts in the two micrographs and have replaced the images in Fig. 4f (previously Fig. 3f) and Extended Data Fig. 4a with the underlying raw widefield micrographs.

Reviewer #4 (Remarks to the Author):

My view is that the paper does make an advance over previous work on the topic. In my opinion, the major potential conceptual advance is that the condensates contain pores that are proposed to allow COPII subunit entry but not COPI, and thereby segregate anterograde carriers from retrograde ones at the ER-Golgi interface. This is based on EM and dextran exclusion data. The data are thus rather indirect on this point, which remains quite speculative. This is especially true if one considers that the pores appear heterogenous in size, commented on by another reviewer. In my opinion, further data is needed to strengthen the point that TFG condensates act as such a filter. Imaging COPII and COPI alongside TFG condensates in cells for example could be done, or biochemical evidence showing segregation of COPII versus COPI complexes by the condensates would also be good. In the absence of such new data, the advance shown in this paper could be seen as incremental, at least in conceptual terms.

We agree with Reviewer 4 that TFG's ability to segregate COPI from COPII coats was only indirectly shown in the previous manuscript version. Their concerns are shared by Reviewer 1 in point 8, to which we responded to in detail.

In brief, we have co-transfected HeLa cells with recombinant TFG as well as Sec23, Sec24, or both Sec23/24 and observe a marked enrichment in all cases. This supports that smaller COPII components, even Sec23-Sec24 heterodimers, can efficiently partition into TFG condensates (Fig. 6a, Extended Data Fig. 13e). This is contrasted by the behavior observed for immunostained endogenous coatomer, which we stripped off Golgi membranes using BFA to probe whether it could enter TFG condensates as well. Strikingly, unlike COPII, the COPI coat protein coatomer is efficiently excluded from TFG condensates (Fig. 6a, Extended Data Fig. 13e).

Two *additional* lines of experiments included in the revised manuscript support TFG's ability to function as a molecular 'sieve' at the ER-Golgi interface: i) TFG condensates formed *in vitro* exclude ribosomes, which was predicted based on the molecular weight cutoff determined using fluorescent dextrans, and by the fact that the ER-Golgi interface is a ribosome-free zone. ii) We obtained thin sections of cells in which TFG had been depleted using siRNA (Fig. 6c). When performing TEM on these sections, we observed a marked increase in the number of vesicles near the ER-Golgi interface compared to control cells, concomitant with a disruption of Golgi membranes, supporting our model that TFG acts as a barrier that holds anterograde carriers (Fig. 6c, Extended Data Fig. 13c).

We would like to again thank all Reviewers for their constructive input and thorough assessment of our dataset. We believe that in this second revision of our manuscript, the suggested experiments have greatly improved our conclusions.

Reviewer #1 (Remarks to the Author):

The revised manuscript is a substantial improvement over the initial submission. The authors have provided stronger evidence that TGF forms 'sponge-like' condensates in vitro, dependent on its phosphorylation state. The evidence in cells is not as compelling however, particularly as the authors continue to analyze condensates that form following over-expression of the protein. Nonetheless, the mapping of phosphosites throughout the cell cycle is a very welcome addition, although further analysis of the T87 residue seems warranted, given its presence specifically in mitotic cells, where condensates do not form. It would be relatively straightforward to express a T87E mutant to determine whether it fails to de-condense properly during mitosis.

We would like to thank the reviewer for acknowledging the improvements to our manuscript. We agree that T87 warrants further experimentation and created TFG-T87E-mEGFP-FLAG (phosphomimetic) and TFG-T87A-mEGFP-FLAG (phospho-dead) variants. The propensity of these constructs to form condensates upon overexpression during interphase and mitosis was analyzed, with the new data reported in Figure 3f. As expected, both variants exhibit condensate formation during interphase, albeit with less and smaller condensates per cell compared to wildtype TFG. This highlights a role for T87 in TFG condensation, putatively due to its contribution to PB1 domain folding and thus homooligomerization of TFG. Strikingly, during mitosis, condensates efficiently dissolved for wildtype TFG (90% clearance) and the phosphomimetic mutant T87E (90% clearance) whereas the phospho-dead mutant T87A reproducibly shows an increased fraction of remnant-containing mitotic cells (65% clearance). These data support that phosphorylation of T87 is indeed important for condensate disassembly, but indicates that likely additional mitotic sites are needed to lead to a full dispersal of the TFG dense phase.

By contrast, the data suggesting that the sponge-like condensates insulate secretory carriers (COPII) from retrograde carriers (COPI) at the ER/Golgi interface continues to remain weak, particularly given how prominently this idea is presented in the study. The thin section EM data are difficult to interpret without immunogold labeling of COPII vs. COPI, and no quantification of the imaging data shown is provided. Without more direct evidence, the conclusion that TFG acts as a molecular sieve to prevent COPI vesicles from invading the space used by COPII seems premature.

We would like to point to Extended Data Figure 13e in our previous submission, which contained a quantification of the EM data. For clarity, we have moved the quantification to the main figure, in which we observe a significant increase in the number of Golgi-associated vesicles upon TFG knockdown (Fig. 6e).

Using immuno-EM, it has long been established that COPII and COPI coats are segregated spatially at the ER-Golgi interface (Duden et al., Cell, 1991; Oprins et al., J. Cell Biol, 1993 ; and Orci et al. Cell, 1997). Our data suggests that TFG is responsible for this segregation, since we observe a consistent inclusion of COPII coat components into TFG condensates formed by overexpression, while COPI coats are excluded (Fig. 6a), supporting that this feature would be preserved in endogenous 300-nm condensates (Fig. 2a). We were not able to resolve a

segregation of coats for endogenously tagged TFG due to the diffraction limit (250 nm in XY) robustly. We agree that ultimately, localizing both coat types in context of TFG at endogenous levels and ideally in context of ERES membranes would provide direct evidence, which we are working towards obtaining via cryo-ET. Therefore, we have toned down our conclusions throughout the manuscript to mirror the fact that our data *suggests* TFG functioning as a molecular 'sieve' as follows:

Line changes:

Line 314: TFG condensates insulate COPII from COPI coats → TFG condensates *putatively* insulate COPII from COPI coats

Line 325: These data support that TFG serves as a barrier at the ER-Golgi interface that insulates carriers and excludes ribosomes. → These data support that TFG *could* serve as a barrier at the ER-Golgi interface that insulates carriers and excludes ribosomes.

Line 330: We find that TFG indeed forms condensates at physiological conditions via hydrophobic 'sticker' residues, and assembles into hollow, porous spheres that act as molecular 'sieves'. → We find that TFG indeed forms condensates at physiological conditions via hydrophobic 'sticker' residues, and assembles into hollow, porous spheres that *may* act as molecular 'sieves'.

Line 360: Altogether, these data support that TFG condensates are indeed able to segregate COPII from COPI coats as predicted from our in vitro data. → Altogether, these data support that *overexpressed* TFG condensates are indeed able to segregate COPII from COPI coats as predicted from our in vitro data.

Line 362: COPII coats can access the lumen of TFG condensates, thus creating both a diffusion-limited space for anterograde transport towards the Golgi, while restricting COPI-mediated retrograde traffic to the periphery of TFG condensates. → COPII coats can access the lumen of TFG condensates, *putatively* creating both a diffusion-limited space for anterograde transport towards the Golgi, while restricting COPI-mediated retrograde traffic to the periphery of TFG condensates.

Line 379-383: TFG condensates acting as molecular 'sieves' would be supported by the exclusion of ribosomes from the ER-Golgi interface and the segregation of vesicle-forming machinery (with COPII coats localizing to the interface, and COPI coats segregated to its periphery), both of which we confirmed experimentally. → TFG condensates *may* act as molecular 'sieves,' which would be supported by the exclusion of ribosomes from the ER-Golgi interface and the segregation of vesicle-forming machinery (with COPII coats localizing to the interface, and COPI coats segregated to its periphery), both of which we confirm in vitro or in vivo for *TFG condensates formed via overexpression*.

Line 632: This results in a spatial segregation of anterograde from retrograde traffic, and the creation of a diffusion-limited space for bidirectional transport at the ER-Golgi interface. → This *putatively explains* a spatial segregation of anterograde from retrograde traffic, and the creation of a diffusion-limited space for bidirectional transport at the ER-Golgi interface.

Minor comment - there is a typo in line 334 ('byd' should be 'by')

We apologize for this typo and have corrected it.

Reviewer #2 (Remarks to the Author):

Thank you to the authors for thoroughly addressing concerns raised by all the reviewers. It is an interesting story likely to be impactful for understanding the ER-golgi interface. The work presents a potential new functional role for condensates in selective transport. There are still a few issues with the manuscript, some of which that arose from the revised figures and additional experiments. The molecular origins of the hollow condensate morphology remain unresolved, but potential biological explanation for the function of these hollow condensates is presented with additional experiments and rationale.

Concerns:

Protein concentration for the in-vitro images shown in Figure 1 (specifically 1E) are not shown or explained in the text. This hinders interpretation of their figures, especially across the main and extended data figures where multiple constructs and concentrations are tested. Does the zoom-in of Figure 1H highlight hollowness of condensates?

We thank Reviewer 2 for their enthusiasm and careful re-consideration of our revised manuscript. The starting protein concentration in Fig 1e is 430 nM, which we added to the figure legend. The zoomed-in Z-sections depicted in Figure 1e highlight the anisotropy of an individual condensate. The micrographs in Figure 1h provide visual correlates for the dilute phase (no condensates, open circles in plot), dense phase (condensates, grayed out circles in plot), and the inverted phase (inverted phase, solid black circles in plot), respectively. The plot corresponds to the protein concentrations shown in the representative visual correlates. For these specific images, the dilute phase (left) derives from protein at a concentration of: < 5 nM; for the dense phase (middle): 200 nM; and a TFG inverted phase (right) at 6.4 μ M. We have added these concentrations to the figure legend. The starting concentration in the evaporation assay (Fig. 1d) is ~ 100 nM.

Have you tried collecting imaging movies to ascertain how coalescence of these droplets contributes towards formation of anisotropic condensates? This would really help address several questions concerning the hollow condensate morphology. Extended data Fig 2CD are useful, but these are different constructs (TFG 162-240-mEGFP-FLAG condensates).

We indeed observe such coalescence during the evaporation test, which shows the formation and fusion of initially isotropic TFG condensates that reorganize into an anisotropic, sponge-like distribution spontaneously after approximately 10 minutes post the initial phase transition (visual correlates provided in Fig. 1d). We apologize for the confusion, but Extended Data Figure 2c,d depict experiments assessing the contribution of nucleic acids to the anisotropic distribution of condensates. *In vivo*, we were not able to determine whether the initial foci observed upon phase transition of overexpressed are anisotropic. While endogenously tagged TFG exhibits anisotropic condensates, we agree that it is indeed an intriguing question whether these result from initially isotropic condensates.

The image in Figure 2D and comparison with Extended Data Fig 6 makes it hard to see whether this is anomalous cell or a cell with very large condensates, especially in comparison to the images shown in Extended Data Figure 4A.

We apologize for the confusion – Fig. 2d indeed depicts magnifications of large individual condensates, and not individual cells, that we frequently observe for wt TFG-mEGFP overexpressing cells. These condensates were subjected to different imaging modalities to assess whether their dense phase exhibited an anisotropic distribution as observed in vitro. We have edited the figure legend in Fig. 2 to make this point clearer.

The addition of the phosphorylation studies has improved the manuscript. With the insights here, it does make this reviewer want to know more about what specific aspects of phosphorylation (and/or other modifications) are critical to the hollow condensate morphology (especially as bacterially-expressed protein does not form hollow condensates). It appears that the degree of phosphorylation coupled with which sites are phosphorylated are important to controlling condensation properties of TFG since hyperphosphorylation appears to inhibit condensate assembly for this protein. It would be important to iron this out for this paper, but I understand the complexities in doing so. The authors do frequently suggest that it is outside the scope of the current work. Readers will want to know the molecular basis for this porous type of condensate (possibly tied to changing material properties over time – see comment above regarding imaging movies).

We fully agree that further biophysical/biochemical characterization of the molecular mechanism, including simulations, are warranted to further characterize this unusual condensation mechanism. In future work, we will assess the contribution of each phosphosite during interphase and mitosis to the propensity of TFG to form and disassemble condensates. Specifically for mitotic phosphosites, placeholder modifications such as O-GlcNacylation could significantly contribute to the molecular mechanism of TFG condensation, providing an explanation for the diminished capability of TFG-T87A to form condensates. We do believe that such further characterization goes beyond the scope of this manuscript, which establishes these novel points:

- i) the ability of TFG to form condensates at physiological conditions
- ii) identification of the anisotropic nature of TFG condensates
- iii) regulation of TFG condensates via phosphorylation
- iv) the contribution of hydrophobic ‘stickers’ to condensation mechanism of TFG
- v) permeability of TFG condensates, functioning as molecular ‘sieves’
- vi) a potential role for TFG condensates in segregating COPII from COPI carriers

The ribosome data in Figure 6B initially is confusing as it appears that ribosomes are recruited to the TFG condensate, as opposed to the exclusion the authors mention in the text. Additional replicates/images are required for this.

We agree with Reviewer 2, and apologize for the confusion. While we observe a robust exclusion of AF647-labeled ribosomes within the lumen of TFG condensates, ribosomes indeed

appear to be attracted to the surface of the condensate. We have therefore revised our labeling strategy by employing the more hydrophobic SiR dye and repeated the experiments. As shown in new Fig. 6b, SiR-labeled ribosomes are not attracted to TFG condensates, and a robust exclusion from the lumen of TFG condensates is observed (Fig. 6b). As indicated for three representative condensates, a clear exclusion is observed for 60% of condensates ($n = 105$), with the remaining 40% representing condensates that are either too small to optically exclude ribosomes present in confocal planes above the condensate, or exhibiting defects in the respective condensate surface. We thus believe that these data support that TFG may be responsible for the long observed ribosome exclusion zone in the Golgi interface, as predicted by its ability to function as a molecular sieve *in vitro*.

They still have different cells in their temperature ramp-down live imaging experiment of figure 4 (4G). They did follow the same cell for the temperature ramp-up experiment, but it appears to be a sick cell, given odd nuclear vacuolar morphology. Are there movies that the authors can supply for these experiments? It is interesting to note that these experiments do not exhibit hollow condensate morphology, but perhaps these do if you are able to image over longer time.

We apologize for the confusion. These images are based on widefield microscopy, which hampers discernment of anisotropy in condensates optically (in contrast to confocal microscopy). Widefield was chosen as a microscopy modality due advantages in short exposure times to capture whole cells, and high sensitivity for live cell, time-lapse imaging. The HeLa CCL2 cells indeed often exhibit fragmented nuclei or unusual morphologies, but we were able to observe this effect robustly for different cells and in independent experiments. In lieu of movies, we are providing galleries for two such examples of temperature ramp-up experiments (providing all time points) as Reviewer Fig. 1 below. We thank Reviewer 2 for raising the interesting point suggesting that temperature, in addition to modulating TFG condensation, may also contribute to the anisotropic distribution in the condensate dense phase. This would be supported by a reduction of sphericity of TFG condensates with increasing temperature as depicted in Fig. 4h, with aspherical condensates coinciding with an anisotropic dense phase (Fig. 2d).

Reviewer Figure 1: Temperature ramp-up experiment of HeLa cells expressing TFG. Time intervals for each panel are 5 min for a total of 2 hr. Temperature starts at 22°C and increases gradually to 37°C. Two examples of individual cells are given.

Reviewer #4 (Remarks to the Author):

The authors have performed additional experiments to address the point I raised about showing COPII and COPI coat segregation. These indicate that COPII subunits can enter condensates made upon TFG over-expression whereas COPI appears excluded. This would support the model whereby TFG spatially segregates ER to Golgi intermediates. However, I have several concerns over the nature of the data. The first relates to point that this is only seen with TFG overexpression. Numerous studies have observed endogenous COPII and COPI adjacent to each other in very close proximity at ERES and ERGIC in unperturbed cells. This does not fit with such a dramatic segregation as is shown upon TFG over-expression (Fig 6a).

We would like to emphasize that we do not propose that our data establishes a segregation of COPI and COPII carriers at the ER-Golgi interface – this has been indeed already established for decades by several groups employing immunogold labeling and electron microscopy (see Reviewer Figure 2; Martinez-Menarguez et al., 1999; Duden et al., Cell, 1991; Oprins et al., J. Cell Biol, 1993; and Orci et al. Cell, 1997), showing a spatial segregation of carriers corresponding to the dimensions of TFG condensates as observed at endogenous levels (Fig. 2a). If TFG is indeed a condensate at the ER-Golgi interface, enlarging it via overexpression would be predicted to emphasize a segregation of COP coats, particularly if TFG is responsible for their segregation in the first place.

Furthermore, we would like to emphasize that overexpressing proteins that give rise to biomolecular condensates in order to assess their ability to recruit client proteins is a wide-spread and established approach, specifically if endogenous condensates are diffraction-limited (Rebane et al., FEBS Lett, 2019; Brangwynne et al., Cell, 2019; Molliex et al., Cell, 2015; Yu et al. NCB, 2021; Xie et al. Nature, 2025). Additionally, the condensates we selected in Fig. 6a for inclusion/exclusion scoring were approx. 1 μm , whereas the native size of TFG condensates is ~ 300 nm. As depicted in new Extended Data. 13f, inclusion of COPII in TFG condensates with a concomitant exclusion of COPI was robustly observed for various condensate diameters, including equal to those of endogenous TFG condensates (Fig. 2a), reflecting that biochemical and biophysical properties are conserved and largely independent of the volume of the condensate dense phase.

Some of the other data in Fig 6 is also not convincing e.g. ribosomes do not appear to be excluded from the TFG condensates shown in Fig 6b, contrary to the author's conclusions.

We indeed observed a robust exclusion of AF647-labeled ribosomes from the lumen of TFG condensates, which was evident from the colocalization and line scan analyses. However, it is correct that AF647-labeled ribosomes were attracted to the condensate surface. We have addressed this by repeating the experiments with SiR-labeled ribosomes (a more hydrophobic dye), and as shown in new Fig. 6b, find that SiR-labeled ribosomes are no longer attracted to TFG condensates, while being robustly excluded from the lumen of TFG condensates (Fig. 6b).

Bogus, Wegeng, et al., unpublished

modified from: Oprins et al., 1993

modified from: Orci et al., 1997

A 200 nm

C 200 nm

Modified from: Martinez-Menarguez et al., 1999

Reviewer Figure 2: Segregation of COPI and COPII coats at the ER-Golgi interface. Top: Overexpressed FLAG-TFG-SNAP condensates in HeLa cells with overexpressed Sec24 inclusion and immunostained β COP exclusion (this manuscript; Fig. 6a). Middle and bottom: Immuno-electron microscopy micrographs depicting COPII coats segregated from COPI coats at the ER-Golgi interface (Oprins et al., 1993; Martinez-Menarguez et al., 1999; Orci et al., 1997, Martinez-Menarguez et al., 1999). Dotted circles indicate the diameter of TFG

condensates (~ 300 nm) determined in vivo and in vitro (this manuscript; Fig. 2a and Fig. 1g, respectively).

The EM data in Fig 6c showing more vesicles in the vicinity of the ER and Golgi could be due to many reasons other than loss of TFG condensates in this region. For example, these could well be Golgi-derived vesicles considering that the Golgi morphology looks quite disrupted upon loss of TFG.

We agree with Reviewer 2 that vesicles in 6c could also be Golgi-derived (COPI). However, they might also represent uncoated COPII carriers that are not contained by a TFG 'cage' any more. Loss of TFG would result in an aberrant diffusion of anterograde carriers, and concomitantly, a reduced rate of cargo flow from the ER to the Golgi, which we indeed observe experimentally (Extended Data Fig. 3c,d), and is further consistent with an aberrant distribution of COPII carriers and ERGIC upon TFG knockdown in *C. elegans* (Witte et al., 2011).

The model shown in Fig 6d showing a relatively large condensate at the ER-Golgi interface needs to be supported by analysis of the endogenous TFG protein in the absence of over-expression. This would appear to be key data that is necessary to avoid the criticism that the observations reflect an artificial situation that does not recapitulate the true organisation of the ER-Golgi interface.

We understand Reviewer 4's concerns, but would like to highlight that the dimensions of the condensate we depicted in the model (former Fig 6d; \approx 300 nm) was indeed based on the size of TFG condensates observed at endogenous expression levels (Fig. 2a), which is further paralleled by the average size of condensates observed in vitro (Fig. 1g). The approximate diameter of ERES stems from TANGO1 micrographs obtained with STED microscopy (Raote et al., 2018).

However, we agree that cis-Golgi membranes were not drawn proportionally, given that the diameter of an individual Golgi ministack is approx. 1 μ m. Therefore, we have edited the model in new Fig. 6e to reflect this fact, and to emphasize the nature of laterally linked cisternae in the Golgi ribbon.

We would once again like to thank all reviewers for their time and constructive and thorough feedback, which we believe has led to great improvements of our dataset over the past revisions, now warranting publication at Nature Communications.

REVIEWERS' COMMENTS

Reviewer #1 (Remarks to the Author):

The authors have addressed some of the concerns raised during this most recent round of review. A nice experiment showing the importance of T87 phosphorylation during mitosis is now included, and the authors have toned down the idea that TGF acts as a molecular sieve, as this is not directly shown. Since demonstration of this latter concept would have been a major conceptual advance for the field (it has already been shown by Qiu et al. that TGF undergoes phase separation to facilitate COPII vesicle trafficking), there is now a concern regarding the significance of this study, but this point is best left to the editors to evaluate.

We thank reviewer 1 for their time, expertise, guidance, and appreciation of data added regarding TFG's mitotic T87 phosphosite. We agree that future work employing high-resolution approaches are required to confirm the presence of a TFG 'sieve' *in situ* that our data strongly support.

While largely focusing on vesicle-containing condensates at the presynapse formed by synapsin and Piccolo, Qiu et al. also investigated phase separation of TFG and its ability to sequester vesicles into condensates formed by overexpression. Phase separation of TFG was observed *in vitro* using bacterial extracts and in presence of crowding agents at concentrations that far exceed cellular levels of TFG (endogenous concentration: 0.9 μM ; Qiu et al.: 40 μM + 3% PEG 8,000), resulting in isotropic condensates.

Since our *in vitro* data is based on recombinant protein obtained from human suspension cells, we were able to identify the anisotropic nature of TFG condensates that form at physiological concentrations and conditions (saturation concentration: 0.1 μM), and importantly, in absence of crowding agents. The hollow condensate nature observed *in vitro* was confirmed by endogenous tagging of TFG *in vivo*. We identify the molecular mechanism behind TFG condensation, which depends on hydrophobic 'sticker' residues. Furthermore, we identify TFG as a phosphoprotein, which becomes hyperphosphorylated during mitosis, leading to disassembly of TFG condensates. TFG condensates function as a molecular 'sieve' *in vitro*, and the predicted cutoff correlates with exclusion of both ribosomes and COPI, but not COPII coats *in vivo*, resulting in our proposed model that TFG condensates spatially organize bidirectional transport in the early secretory pathway.

We believe these and other data provided in our manuscript provide significant advances to our understanding of ER-Golgi interface and thank the editorial team for their support of our manuscript.

Reviewer #2 (Remarks to the Author):

I thank the authors for clarifying points and modifying figures in this revised manuscript. I have no additional concerns.

We thank Reviewer 2 for their time, expertise, and highly constructive guidance throughout the revision process, which we believe has greatly strengthened the manuscript.

Reviewer #4 (Remarks to the Author):

The authors have addressed my points in their new revised version of the manuscript. I would be happy to support publication.

We thank Reviewer 4 for their time, expertise, highly constructive feedback, and enthusiastic support of our revised manuscript.